# Dissecting regulatory pathways for transcription recovery following DNA damage reveals a non-canonical function of the histone chaperone HIRA

Déborah Bouvier[1], Juliette Ferrand[1], Odile Chevallier[1], Michelle T. Paulsen[2], Mats Ljungman[2] & Sophie E. Polo [1][✉]

Transcription restart after a genotoxic challenge is a fundamental yet poorly understood process. Here, we dissect the interplay between transcription and chromatin restoration after DNA damage by focusing on the human histone chaperone complex HIRA, which is required for transcription recovery post UV. We demonstrate that HIRA is recruited to UV-damaged chromatin via the ubiquitin-dependent segregase VCP to deposit new H3.3 histones. However, this local activity of HIRA is dispensable for transcription recovery. Instead, we reveal a genome-wide function of HIRA in transcription restart that is independent of new H3.3 and not restricted to UV-damaged loci. HIRA coordinates with ASF1B to control transcription restart by two independent pathways: by stabilising the associated subunit UBN2 and by reducing the expression of the transcription repressor ATF3. Thus, HIRA primes UV-damaged chromatin for transcription restart at least in part by relieving transcription inhibition rather than by depositing new H3.3 as an activating bookmark.

[1] Epigenetics & Cell Fate Centre, UMR7216 CNRS, Université de Paris, Paris, France. [2] Department of Radiation Oncology, University of Michigan, Ann Arbor, MI, USA. [✉]email: sophie.polo@univ-paris-diderot.fr

The maintenance of gene expression programmes established during organism development is critical for preserving cell functions and identities[1]. Yet, modulations in transcriptional programmes are indispensable for the cellular response to environmental cues, including the response to genotoxic stress. DNA damage indeed elicits substantial rewiring of transcription, with the induction and repression of damage-responsive genes[2]. DNA double-strand breaks lead to the local inhibition of ongoing transcription through damage signalling pathways and to the production of small non-coding RNAs[3–5], while other DNA lesions, including UV photoproducts, physically block the progression of elongating RNAPII[6–8]. Moreover, following the rapid and local inhibition of elongating RNAPII at sites of UV damage, UV irradiation triggers global repression of transcription initiation, which also affects undamaged genes[9–14]. The repair of transcription-blocking UV lesions by the dedicated Transcription-Coupled Repair (TCR) machinery, a subpathway of Nucleotide Excision Repair (NER)[15,16], is necessary for the recovery of transcriptional activity[17]. TCR involves RNAPII backtracking, release from its template, and also ubiquitylation and proteasomal degradation in case of persistent stalling[18–22]. The ubiquitin-dependent degradation of RNAPII is a major determinant in transcription regulation following UV damage[21,22]. The release of UV–stalled RNAPII entails eviction of ubiquitylated RNAPII from chromatin by the Valosin-Containing Protein (VCP) segregase[23,24], which targets RNAPII to proteosomal degradation in yeast and human cells[25,26]. Importantly, failure to recover from UV-induced transcription inhibition has deleterious consequences including cell death and disease. It is indeed one of the main features of Cockayne Syndrome[17]. However, the molecular mechanisms underlying transcription restart after UVC damage repair are still poorly understood. Several transcriptional regulators have been involved in this process[9,27–35] in addition to core TCR components and factors modulating their stability[36], but how they possibly cooperate in controlling transcription restart after UVC damage remains elusive. In line with the central role of chromatin dynamics in transcriptional regulation[37], chromatin changes associated with the DNA damage response are of functional importance for restoring transcriptional activity[38]. In particular, several histone chaperones[39] and modifying enzymes[40] were shown to promote transcription recovery following UVC damage in mammalian cells, including the H3K79 methyltransferase DOT1L[28], and the core histone chaperones Facilitate Chromatin Transcription (FACT)[30] and Histone Regulator A (HIRA)[29,41].

HIRA is an evolutionarily conserved histone chaperone complex that promotes the deposition of H3.3 histones on DNA in a replication-independent manner[42–45]. Interestingly, HIRA activity is intimately linked to transcriptional regulation as this chaperone deposits H3.3 at gene bodies and regulatory elements such as promoters and enhancers in mammalian cells[46,47]. In response to UVC irradiation, HIRA is recruited to damaged chromatin regions in a ubiquitin-dependent manner by a mechanism that is not yet fully elucidated, and deposits newly synthesised H3.3 histones[29]. Based on the observation that HIRA was only transiently recruited to damage sites and released long before transcription restart, we proposed a bookmarking mechanism whereby this histone chaperone was priming chromatin for transcription recovery[29]. An attractive candidate for the activating bookmark was H3.3 because of the strong connection between the deposition of this histone variant by HIRA and transcriptional activation. Indeed, HIRA-mediated H3.3 deposition plays a critical role in transcriptional reprogramming in Xenopus[48] and mouse cells[49], and is key for reaching the full dynamic range of transcription in mouse developing oocytes[50]. Furthermore, unscheduled H3.3 deposition by HIRA governs a pro-metastatic transcriptional programme in breast cancer cells[51]. HIRA-mediated H3.3 deposition is also required for ribosomal gene transcription in the mouse zygote[52], for myogenic gene transcription in mouse myoblasts[53], for the transcription of angiogenic genes in endothelial cells[54], and for viral gene expression[55]. However, besides these multiple links with active transcription, H3.3 deposition by HIRA is also required for the establishment of silencing at developmental genes in mouse embryonic stem cells[56], raising questions about the contribution of H3.3 to HIRA function in transcription recovery after UVC damage repair. Here, by dissecting the mechanisms of HIRA-mediated transcriptional regulation following UVC damage in human cells, we rule out the contribution of new H3.3 deposition to transcription restart and uncover a non-canonical function of HIRA in this process that promotes transcription recovery on a genome-wide scale.

## Results

**HIRA function in transcription restart after UVC damage is independent of new H3.3 deposition.** As a first step towards dissecting the mechanisms for HIRA-mediated transcription regulation in response to genotoxic stress, we sought to narrow down which subunit of the HIRA complex was important for transcription recovery after UVC damage repair in human cells. Indeed, this histone chaperone complex comprises HIRA[57], Ubinuclein 1 (UBN1)[58] or the less characterised UBN2[59], and Calcineurin-binding protein 1 (CABIN1) subunits[60] (Fig. 1a), and coordinates with Anti-silencing function 1 A (ASF1A)[43,61] to promote H3.3 deposition (reviewed in[42]). Central to the complex, HIRA serves as a scaffold, stabilising the other core subunits. Consistent with this, we observed that HIRA knock-down in HeLa and U2OS cells led to a strong down-regulation of UBN1 and CABIN1 while ASF1A levels remained unaffected. In contrast, knocking down UBN1 or CABIN1 did not have a marked impact on HIRA levels (western blot panels on Fig. 1a, b).

We thus examined transcription regulation after UVC damage by labelling nascent transcripts with 5-ethynyluridine (EU) in cells knocked down for individual subunits of this histone chaperone complex (Supplementary Fig. 1a). As defined previously[29], we analysed transcription inhibition and recovery 2 h and 24 h after UV irradiation, respectively. Remarkably, we observed that HIRA down-regulation, and not that of UBN1 and CABIN1, strongly impaired transcription recovery 24 h after UVC damage (Fig. 1a, see Supplementary Fig. 1b-c for raw transcription data in undamaged conditions and for HIRA complex subunit levels at the end of the experiment). The effect of HIRA knock-down was not explained by major alterations in cell cycle distribution and UV-induced cell cycle arrest (Supplementary Fig. 1d), consistent with previous observations[29]. In contrast with transcription recovery that is impaired only upon HIRA depletion, new H3.3 deposition at UVC damage sites (Cyclobutane Pyrimidine Dimers, CPD) was reduced to a comparable extent upon knock-down of HIRA, UBN1 or CABIN1 (Fig. 1b; see Supplementary Fig. 1e for a scheme of the new H3.3 deposition assay). Similar results were obtained by knocking down HIRA complex subunits with a second set of siRNAs and by labelling nascent transcripts with bromouridine (BrU) (Supplementary Fig. 1f). Thus, the ability to restart transcription following UVC damage correlates with HIRA levels but not with new H3.3 deposition at UV damage sites. These findings strongly suggest that the de novo deposition of H3.3 is not critical for transcription recovery after repair of UVC damage. Moreover, UBN1 was recently shown to determine H3.3-specific binding in the HIRA complex[62], but is not required for transcription restart (Fig. 1a and Supplementary Fig. 1f). To directly test the

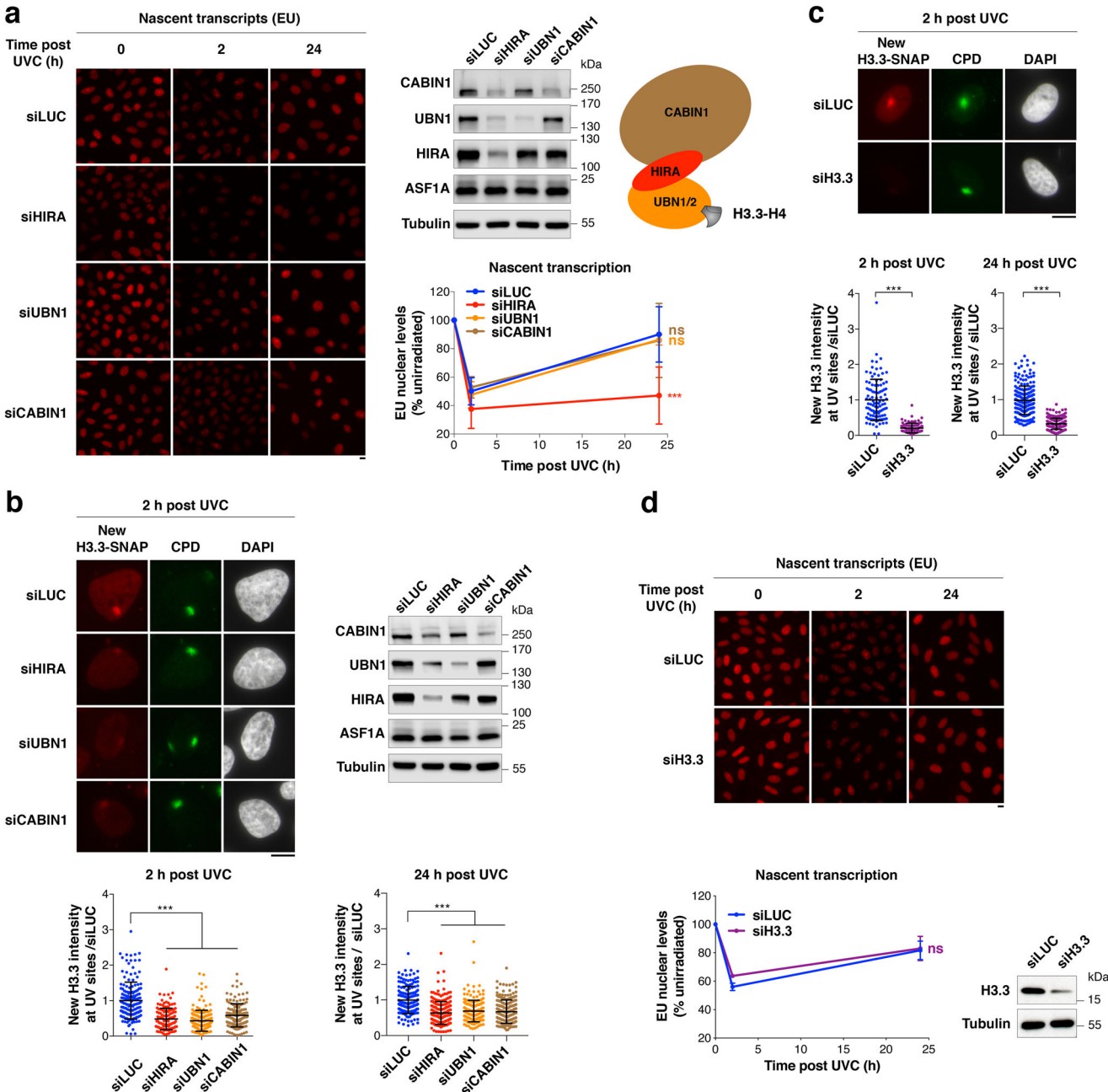

**Fig. 1 HIRA-mediated transcription restart post UVC damage is independent of new H3.3 deposition. a** Fluorescence images showing nascent transcripts labelled with ethynyl-uridine (EU) in unirradiated (0 h) or UVC-irradiated (2 h, 24 h) HeLa cells treated with the indicated siRNAs (siLUC, control). Knock-down efficiencies are controlled by western blot at the time of UV irradiation (Tubulin, loading control). The scheme on the right represents the HIRA complex chaperoning H3.3-H4. The graph shows quantification of nascent transcript levels post UVC relative to unirradiated cells using the same colour code as the scheme for HIRA complex subunits. **b, c** New H3.3 accumulation at UVC damage sites (marked by CPD immunostaining) analysed 2 h and 24 h post local UVC irradiation in U2OS H3.3-SNAP cells treated with the indicated siRNAs (siLUC, control). The scatter plots show new H3.3 levels at UV sites (mean ± s.d. from at least 86 cells scored in three independent experiments). siRNA efficiencies are controlled by western blot for HIRA complex subunits (Tubulin, loading control), and by new H3.3-SNAP labelling for H3.3. **d** Nascent transcript levels analysed by ethynyl-uridine incorporation (EU) as in (**a**). The line graphs show mean values ± s.d. from four (**a**) and three (**d**) independent experiments scoring at least 81 cells per experiment. Statistical significance is calculated by two-way (**a, d**) and one-way ANOVA (**b**) with Bonferroni posttest, and via two-sided Student's *t*-test with Welch's correction (**c**). ***: $p < 0.001$; ns: $p > 0.05$. Scale bars, 10 μm. Source data are provided as a Source Data file.

importance of the H3.3-chaperone activity of the complex for transcription recovery post UVC damage, we targeted H3.3 by RNA interference thus impairing new H3.3 protein production. This resulted in strongly reduced levels of newly deposited H3.3 at UV sites as expected, but did not affect transcription regulation following UVC irradiation (Fig. 1c, d). Collectively, these data

establish that HIRA promotes transcription recovery independently of new H3.3 deposition.

**HIRA does not contribute to transcription-coupled DNA repair.** Having ruled out the contribution of new H3.3 histone deposition to HIRA-dependent transcription restart post UVC,

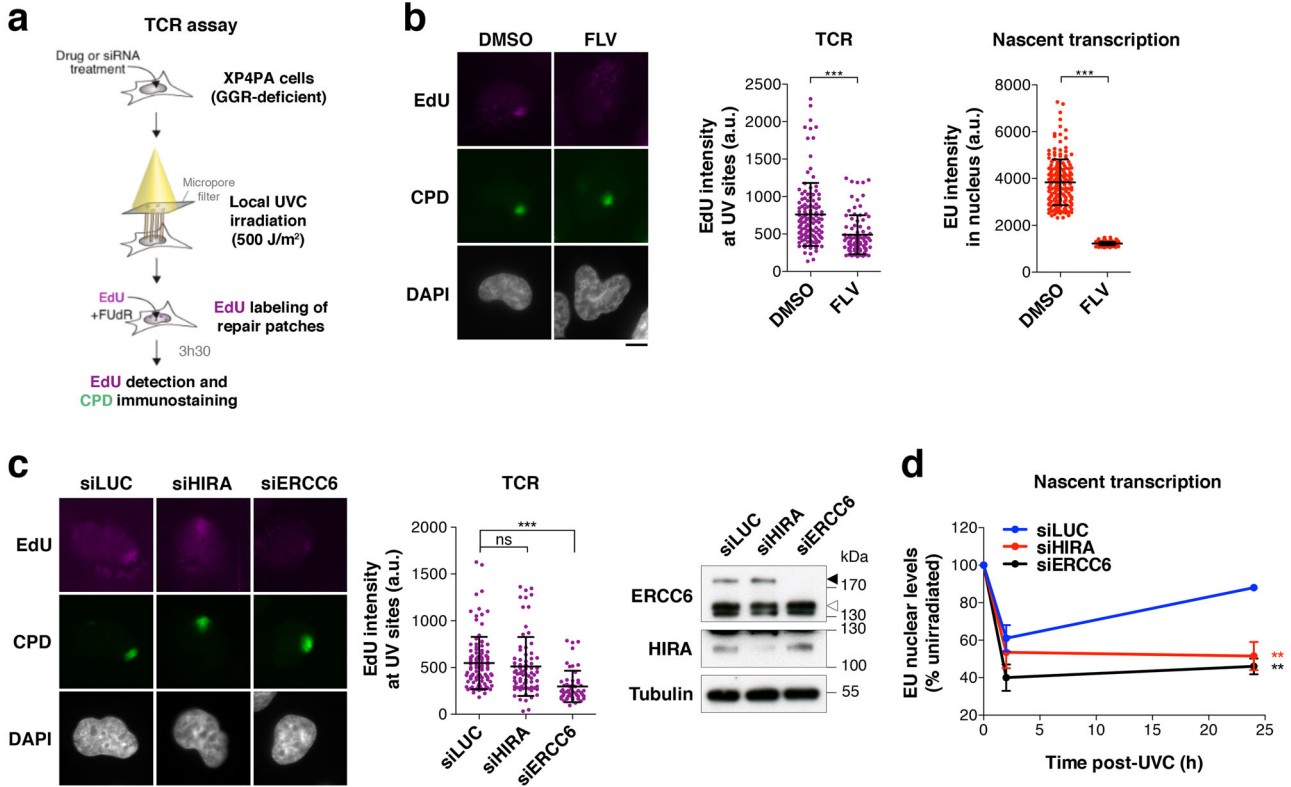

**Fig. 2 HIRA does not contribute to Transcription-Coupled Repair. a** Schematic representation of the Transcription-Coupled Repair (TCR) assay performed in GGR-deficient fibroblasts (XPC-null). **b–c** TCR assay in cells treated with (**b**) the transcription inhibitor flavopiridol (FLV; DMSO, vehicle) or (**c**) the indicated siRNAs (siLUC, control). EdU marks repair synthesis. Knockdown efficiencies are controlled by western blot (black arrowhead, full-length ERCC6; white arrowhead, ERCC6 splice variant; Tubulin, loading control). Transcriptional activity in each condition is monitored by EU staining of nascent transcripts. The scatter plots show mean values ± s.d. from at least 53 UV damage sites scored in a representative experiment out of three. a.u. arbitrary units. Scale bars, 10 μm. **d** Nascent transcript levels analysed by EU incorporation in UVC-irradiated GGR-deficient fibroblasts treated with the indicated siRNAs (siLUC, control). The line graph shows mean values ± s.d. from two independent experiments (at least 50 cells were scored in each condition). Statistical significance is calculated by two-sided Student's $t$-test (**b**), one-way (**c**) and two-way ANOVA (**d**) with Bonferroni posttest. **: $p < 0.01$; ***: $p < 0.001$; ns: $p > 0.05$. Source data are provided as a Source Data file.

we searched for other local events at UV-damaged loci that could mediate this response. We first examined whether HIRA would contribute to TCR of UV lesions, as recently shown for another histone chaperone, FACT[63]. For this, we used fibroblasts that only rely on TCR to repair UV lesions because they are deficient in the other NER subpathway, Global Genome Repair (GGR), and we measured repair synthesis by 5-ethynyl-2'-deoxyuridine (EdU) incorporation at sites of local UVC irradiation[35] (Fig. 2a). To confirm the transcription-coupled nature of UV damage repair in this system, we treated cells with a transcription inhibitor (flavopiridol, FLV) and we also knocked down the core TCR factor Excision repair cross-complementation group 6 (ERCC6, also known as Cockayne Syndrome B, CSB), both resulting in a marked inhibition of repair synthesis (Fig. 2b, c). HIRA depletion in contrast, although resulting in defective transcription recovery in these cells (Fig. 2d), did not lead to a significant reduction in repair synthesis activity (Fig. 2c), arguing that HIRA is not a core TCR component. Thus, we conclude that HIRA function in transcription recovery following UVC damage cannot be attributed to stimulation of TCR.

**HIRA is recruited to UV damaged chromatin by the ubiquitin-dependent segregase VCP**. We next focused our attention on the VCP segregase, which promotes the degradation of UV-stalled RNAPII[25,26], a critical step in transcription restart[21,22], and

which, like HIRA[29], is recruited to UVC-damaged chromatin in a ubiquitin-dependent manner in human cells[64]. We thus wondered whether HIRA may crosstalk with VCP to regulate transcription recovery post UVC. Loss-of-function experiments revealed that HIRA was dispensable for VCP recruitment to UVC damage sites (Supplementary Fig. 2a). In contrast, both siRNA-mediated knock-down (Fig. 3a) and chemical inhibition of VCP (with the small molecule inhibitor NMS-873, Fig. 3b) prevented HIRA accrual to UVC-damaged chromatin, while HIRA total levels remained unaffected as measured by western blot in total cell extracts (Fig. 3a, b). In line with these findings, new H3.3 deposition at sites of UVC damage was also dependent on VCP (Supplementary Fig. 2b) and on the GGR factor DNA damage-binding protein 2 (DDB2) (Supplementary Fig. 2c), known to recruit both HIRA[29] and VCP[64] to UVC damage sites. These results establish that VCP drives HIRA recruitment to UVC-damaged chromatin.

**HIRA function in transcription restart after UVC damage is independent of VCP-mediated RNAPII degradation**. We thus tested whether HIRA would cooperate with VCP to promote RNAPII degradation following UVC irradiation. For this, we analysed the UV-induced degradation of elongating RNAPII by western blot[65], focusing on the Serine2-phosphorylated form of RNAPII large subunit, which marks the elongation state. The

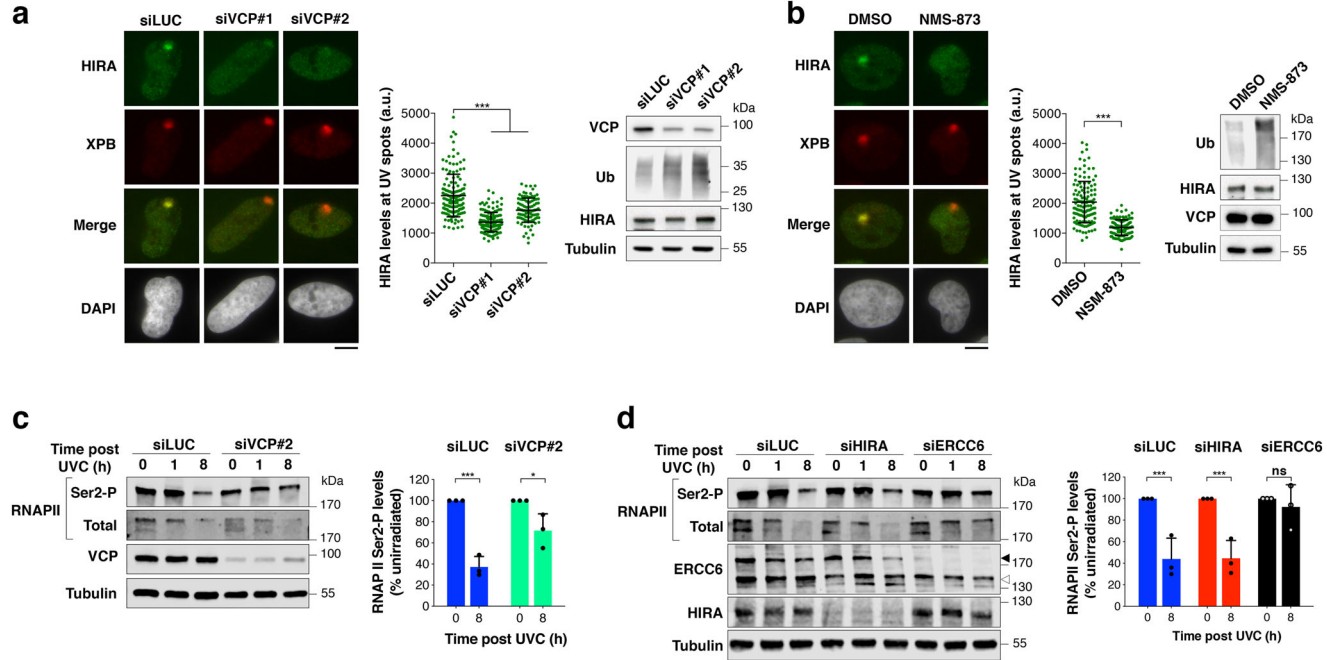

**Fig. 3 Functional interplay between HIRA and VCP at UV-damaged loci. a**, **b** HIRA recruitment to UVC damage sites analysed by immunostaining 20 min after local UVC irradiation in U2OS cells treated with (**a**) the indicated siRNAs (siLUC, control) or with (**b**) the VCP inhibitor NMS-873 (DMSO, vehicle). VCP loss-of-function is controlled by western blot, with an accumulation of ubiquitylated proteins (Tubulin, loading control). Each scatter plot shows data from a representative experiment (mean values ± s.d. from at least 95 cells). Similar results were obtained in three independent experiments. a.u. arbitrary units. Scale bars, 10 μm. **c**, **d** Levels of elongating RNAPII (RNAPII Ser2-P) monitored by western blot at the indicated time points post UVC irradiation in HeLa cells treated with the indicated siRNAs (siLUC, control; black arrowhead, ERCC6 full-length; white arrowhead, ERCC6 splice variant; Tubulin, loading control). Quantification of the decrease in RNAPII Ser2-P levels 8 h post UVC is shown on the graphs. Total protein stain was used for normalisation and results are shown relative to unirradiated cells (mean values ± s.d. from three independent experiments). Statistical significance is calculated by one-way ANOVA with Bonferroni posttest (**a**, **c**, **d**) and two-sided Student's *t*-test (**b**). *: $p < 0.05$; ***: $p < 0.001$; ns: $p > 0.05$. Source data are provided as a Source Data file.

levels of elongating RNAPII dropped 8 h after UVC damage in control cells (siLUC), which was impaired in VCP-depleted cells (siVCP) and abolished in Transcription-Coupled Repair-deficient cells (siERCC6) as expected[26,66] (Fig. 3c, d). In contrast, HIRA knock-down did not significantly inhibit the UV-induced loss of elongating RNAPII (Fig. 3d). These results indicate that the degradation of stalled elongating RNAPII is not sufficient for transcription recovery post UVC damage because RNAPII degradation operates normally in HIRA-knocked down cells, which are defective in transcription recovery. Furthermore, these findings demonstrate that HIRA contributes to transcription restart independently of VCP-mediated degradation of UV-stalled RNAPII.

**Genome-wide impact of HIRA on transcription regulation after UVC damage.** HIRA function in transcription recovery after UV irradiation being unexplained by a local action of HIRA at UV damage sites, we envisioned that HIRA might regulate transcription after UV damage on a genome-wide scale. Thus, we characterised nascent RNA production by deep sequencing of 5-bromouridine-labelled transcripts in cells exposed to global UVC irradiation[67,68] (Bru-seq, Fig. 4a) upon HIRA knock-down by siRNA (Supplementary Fig. 3a, Supplementary Data 1). To take into account differences in global transcription levels between samples, which is a major caveat in this analysis, we performed biological scaling normalisation[69]. For this, we independently measured mean nascent RNA levels per nucleus in each experimental condition relative to the corresponding non-irradiated sample and used those as correction factors (Supplementary

Fig. 3b; see methods for details). Thus, we could identify both UV-repressed and UV-induced genes, including expected candidates among which the immediate early gene *JUN*[70] (Supplementary Fig. 3c). Further underlining the robustness of our analysis, induced genes were overall much shorter (Supplementary Fig. 3c), as reported[71]. We then focused our analysis on UV-repressed genes, which we defined as genes whose transcription was repressed at least 2-fold 2h30 after UVC irradiation in control conditions, and we obtained reproducible results in three independent experiments (Fig. 4b). When assessing the impact of HIRA knock-down on transcription repression and recovery, we observed that HIRA was dispensable for transcription inhibition 2h30 post UV but required for transcription recovery 24h30 post irradiation (Fig. 4c), which is consistent with previous findings[29]. The distribution of Bru-seq reads showed a similar profile along the genes in control and HIRA-depleted cells, with an accumulation of reads in promoter-proximal regions early after UV followed by release from promoter-proximal pausing, as exemplified on the *NIPBL* gene (Supplementary Fig. 4a). Nevertheless, at late time points post UV, transcript levels did not reach back pre-irradiation levels in HIRA-depleted cells (Supplementary Fig. 4a). These results demonstrate that HIRA is not involved in release from promoter-proximal pausing but plays a downstream role in transcription recovery. Importantly, we observed that the vast majority of UV-repressed genes (over 88%) required HIRA for transcription recovery (Fig. 4c, d). HIRA was required for transcription restart independently of gene length, expression level and coding potential (Fig. 4e). Similar results were obtained with another siRNA against HIRA and also when knocking down

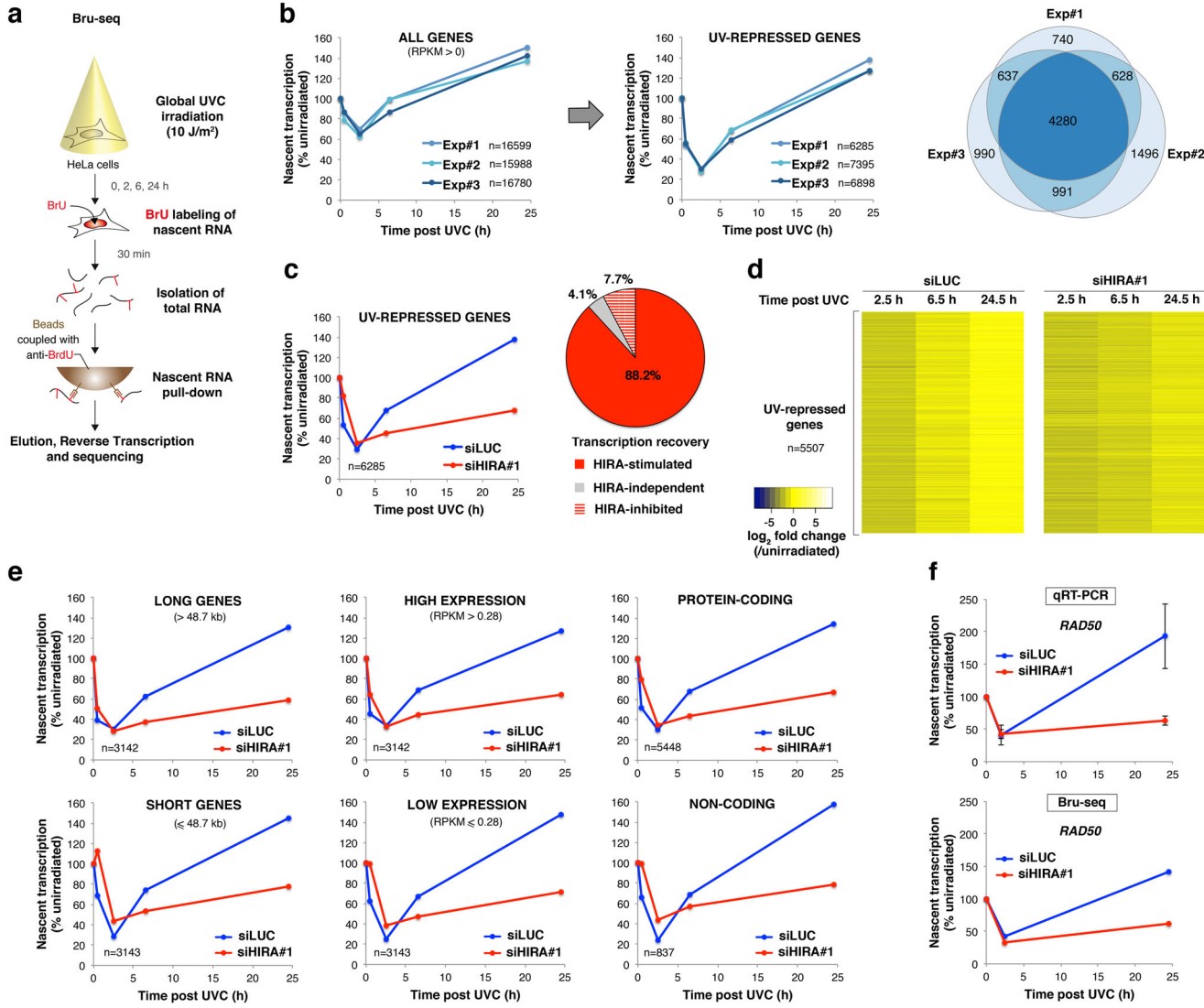

**Fig. 4 HIRA promotes transcription recovery at UV-repressed genes genome-wide. a** Schematic representation of the Bru-seq procedure. **b** Genome-wide analysis of nascent transcription post UVC irradiation in HeLa cells by Bru-seq with biological scaling normalisation (three independent experiments, siLUC treatment). Only genes with detectable transcription in control conditions (RPKM > 0) were analysed. Nascent transcript levels post UVC irradiation normalised to before damage and averaged over n genes are presented on the graphs. The Venn diagram shows the overlap between UV-repressed genes analysed in three independent experiments. **c** Nascent transcript levels post UVC irradiation normalised to before damage in HeLa cells treated with the indicated siRNAs (siLUC, control). Only UV-repressed genes are represented. The pie chart shows UV-repressed genes as a function of their dependency on HIRA for transcription recovery. **d** Heatmap of the data shown in (**c**). Each lane represents one gene. Genes with RPKM = 0 in one of the conditions (n = 778) were excluded. **e** Nascent transcript levels post UVC irradiation normalised to before damage as in (**c**). Long and short genes were discriminated based on median gene length (48,667 bp). Cutoff for low and high gene expression is the median RPKM (0.279766). The distinction between protein-coding and non-coding genes is based on Ensembl and Refseq gene annotations. n number of genes. **f** Validation of Bru-seq results by qRT-PCR on BrU-labelled nascent RNA, focusing on the *RAD50* gene product in HeLa cells treated with the indicated siRNAs (siLUC, control). Mean ± s.e.m. from three independent experiments. Source data are provided as a Source Data file.

the TCR factor ERCC6 (Supplementary Fig. 4b, Supplementary Data 2 and 3). Furthermore, comparison between datasets of genes with defective transcription recovery revealed a very strong overlap between both HIRA knock-downs and also with ERCC6 knock-down (Supplementary Fig. 4c), suggesting that ERCC6 and HIRA might control transcription recovery through shared mechanisms. Regarding the small fraction of escapees that recovered better in the absence of HIRA (HIRA-inhibited: 7.7%, Fig. 4c), gene ontology analyses[72] revealed an over-representation of genes involved in chromatin organisation, mostly canonical histone genes (Supplementary Fig. 4d). In line with these findings, HIRA acts as a transcriptional repressor of histone genes in human cells[73,74], as originally shown for its yeast counterparts[75].

We validated Bru-seq results by quantitative RT-PCR as an orthogonal approach, on BrU-labelled nascent transcripts (Fig. 4f) or on total RNA (Supplementary Fig. 4e), focusing our analysis on short half-life transcripts[76] so that mRNA levels would reflect nascent RNA production. Altogether, these findings demonstrate that HIRA has a large-scale impact on transcription recovery following UVC irradiation.

**HIRA controls UV lesion-independent transcription recovery post irradiation.** Considering that transcription repression post UVC also affects undamaged genes[13,14], we next sought to examine the impact of HIRA on the transcription recovery of

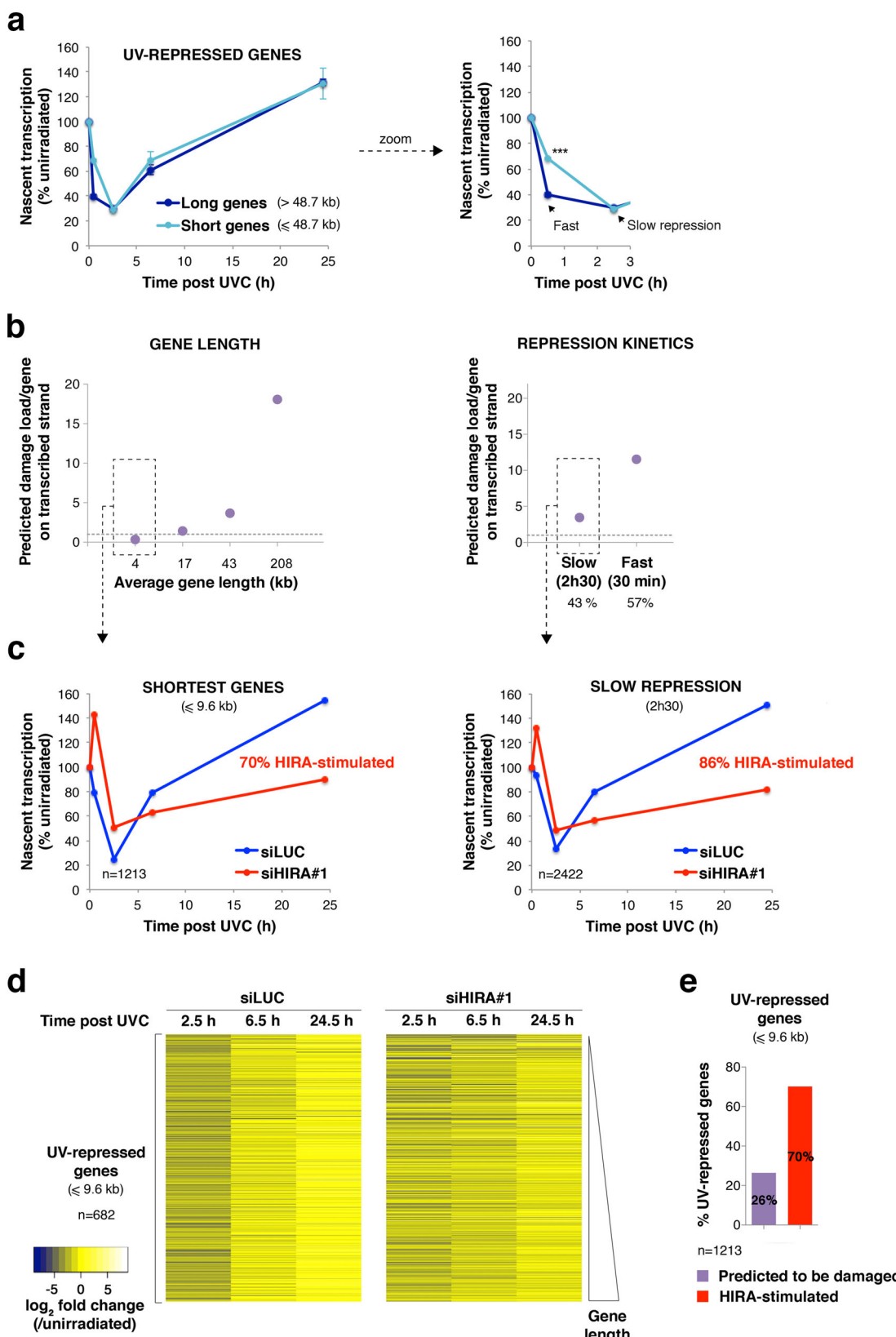

those genes. For this, we partitioned UV-repressed genes according to their length (Fig. 5a), which is a major determinant of the UV damage load. Indeed, UV lesions are distributed throughout the human genome with very few detectable damage hotspots[77–79], thus the UV damage load is generally proportional to gene length and UV dose. Considering that 10 J/m² UVC

irradiation generates 1 UV lesion/6 kb on double-stranded DNA[80], the expected damage load on the transcribed strand, where UV lesions block RNAPII progression, is around 1 UV lesion/12 kb. We observed that in long genes transcription repression was detectable already 30 min post UVC, while short genes generally showed delayed repression (Fig. 5a), consistent

**Fig. 5 The impact of HIRA on transcription recovery is not restricted to UV-damaged genes. a** Nascent transcript levels post UVC irradiation normalised to before damage analysed by Bru-seq in HeLa cells focusing on UV-repressed genes (mean values ± s.d. from three independent experiments, siLUC treatment). Long and short genes were discriminated based on median gene length (48,667 bp). The right graph shows a zoom on early time points. Statistical significance is calculated by two-way ANOVA with Bonferroni posttest (***: $p < 0.001$). **b** Average damage load per UV-repressed gene estimated based on gene length considering that 10 J/m$^2$ UVC generates 1 lesion/12 kb on the transcribed strand (dotted line = 1 UV lesion/gene). UV-repressed genes were partitioned into four quartiles according to their length (left) or into two categories based on repression kinetics (right). Fast repression was defined as at least 2-fold transcription inhibition detectable already 30 min post UVC, while slow repression was observed only 2h30 post UVC. Fast and slow repressed genes represent 57% and 43% of UV-repressed genes, respectively. Mean data from three independent Bru-seq experiments (siLUC treatment). **c** Nascent transcript levels post UVC irradiation normalised to before damage in HeLa cells treated with the indicated siRNAs (siLUC, control) focusing on the indicated subpopulations of UV-repressed genes. Data are from Bru-seq experiment #1. **d** Heatmap of the data shown in (**c**, left panel) focusing on short genes (≤9.6 kb), ranked by gene length. Genes with RPKM = 0 in one of the conditions ($n = 531$) were removed from the analysis. **e** Chart showing the fraction of genes that require HIRA for transcription recovery vs. the fraction of genes expected to be damaged among UV-repressed genes shorter than 9.6 kb. Data are from Bru-seq experiment #1. Source data are provided as a Source Data file.

with an indirect inhibitory effect of DNA lesions on transcription in trans. To assess the impact of HIRA on UV lesion-independent transcription regulation, we thus decided to focus on the shortest genes and on genes that display slow transcriptional repression, i.e. genes that are less likely to harbour UV lesions (Fig. 5b). In both cases, we noticed a strong requirement of HIRA for transcription recovery (Fig. 5c, d). Furthermore, the number of genes that needed HIRA for transcription restart largely exceeded those expected to be damaged. Indeed, based on a Poisson distribution, considering 1 UV lesion/12 kb on the transcribed strand, 26% of the shortest genes are predicted to be damaged on the transcribed strand while HIRA is required for transcription restart in 70% of these genes (Fig. 5e). Noteworthy, HIRA knock-down not only impairs transcription recovery but also mitigates transcription repression of short genes (Fig. 5c).

Altogether, these findings demonstrate that HIRA has a genome-wide impact on transcription recovery following UVC irradiation, and suggest that this is not restricted to genes bearing UV lesions.

**HIRA inhibits the expression of the transcription repressor ATF3.** We next sought to characterise the molecular mechanisms underlying the global effect of HIRA on transcription recovery following UV damage. Recent studies identified the transcription repressor ATF3 (Activating Transcription Factor 3) as a global regulator of transcription post UVC in human fibroblasts, controlling transcription recovery of thousands of genes[31,81]. ATF3 expression is upregulated in response to genotoxic stress and helps maintain UV-induced transcriptional silencing. Supporting the idea that ATF3 mediates a global response to local damage infliction, we observed that ATF3 was upregulated in entire nuclei following local UVC irradiation and did not display any detectable accumulation at UV damage sites (Supplementary Fig. 5a-b). ATF3 levels are known to be controlled by ERCC6, which represses *ATF3* transcription[82] and promotes ATF3 protein degradation post UVC, thus relieving transcription repression of ATF3-target genes[31]. Therefore, we tested if, like ERCC6, HIRA could regulate ATF3 levels. We first examined ATF3 protein levels in total extracts from UV-irradiated cells. The increase in ATF3 levels, detectable 8 h after UVC irradiation, was followed by a decrease in control cells, which was not observed in ERCC6-deficient cells as reported[31,81] (Fig. 6a). Interestingly, HIRA knock-down also led to a continuous increase of ATF3 levels long-term after UVC irradiation, and this without any measurable reduction of ERCC6 levels (Fig. 6a). We obtained similar results when measuring ATF3 nuclear levels by immunofluorescence with a different antibody and by using two distinct siRNA sequences targeting HIRA (Supplementary Fig. 5c). Simultaneous depletion of HIRA and ERCC6 did not result in additive effects on ATF3 levels 24 h after irradiation (Fig. 6b and

Supplementary Fig. 5d), suggesting that HIRA and ERCC6 operate in the same pathway for ATF3 regulation. In line with these findings, the accumulation of ATF3 in HIRA-knocked down cells was not only observed at the protein level but also at the transcript level (Fig. 6c), as reported upon ERCC6 depletion[82]. Supporting a direct role for HIRA in regulating *ATF3* transcription, HIRA occupied regulatory sequences of the *ATF3* gene in HeLa cells[83] (Supplementary Fig. 5e).

Remarkably, the sustained accumulation of ATF3 protein 24 h post UVC was not observed upon knockdown of UBN1 or CABIN1 (Fig. 6d; this is also confirmed for ATF3 transcript in UBN1-depleted cells, Fig. 6c), arguing that the regulation of ATF3 levels does not involve the histone chaperone activity of the HIRA complex. In addition to binding core components of the complex, HIRA also exerts nuclear functions by associating with other H3 chaperones, ASF1A, and to a lesser extent ASF1B[43,84,85]. Interestingly, we found that ASF1B, and not the preferred interacting partner ASF1A, was upregulated after UVC damage (Supplementary Fig. 5f). Furthermore, siRNA-mediated depletion of ASF1B, and not of ASF1A, recapitulated the effect of HIRA knock-down on ATF3 transcript and protein levels post UVC (Fig. 6e, f) and on transcription restart after UV damage (Fig. 6g, Supplementary Fig 5g). Similar to HIRA knock-down, ASF1B down-regulation did not significantly affect cell cycle distribution and UV-induced cell cycle arrest (Supplementary Fig. 1d). We also verified that HIRA levels were unaffected by ASF1B knock-down (Supplementary Fig. 5g) and reciprocally that HIRA knock-down did not lead to loss of ASF1B (Supplementary Fig. 1d).

Altogether, these findings establish that HIRA coordinates with ASF1B to keep ATF3 expression in check post UVC, independently of the histone chaperone activity of the HIRA complex.

**HIRA promotes transcription recovery after UVC damage through ATF3 regulation.** A large fraction of UV-repressed genes (62%), in particular of those that required HIRA for transcription recovery (64%), were found to be targeted by ATF3 post UVC in a previous study[31], including *RAD50* and *NIPBL* (Supplementary Fig. 5h). These data are consistent with a possible role of ATF3 in HIRA-mediated transcription restart. Supporting this idea, we observed an anti-correlation between ATF3 levels and transcriptional activity 24 h post UVC by co-staining for ATF3 and BrU in HIRA-depleted cells (Supplementary Fig. 5i). We thus tested the possibility that HIRA global impact on transcription recovery following UVC damage may be mediated by the reduction in ATF3 levels. For this, we depleted ATF3 in HIRA-knocked down cells and examined if this relieved the transcription defect. Compared to HIRA single knock-down, simultaneous depletion of ATF3 and HIRA brought ATF3 protein down to control levels and partly restored transcription 24 h after UV damage, as measured by BrU-labelling of nascent

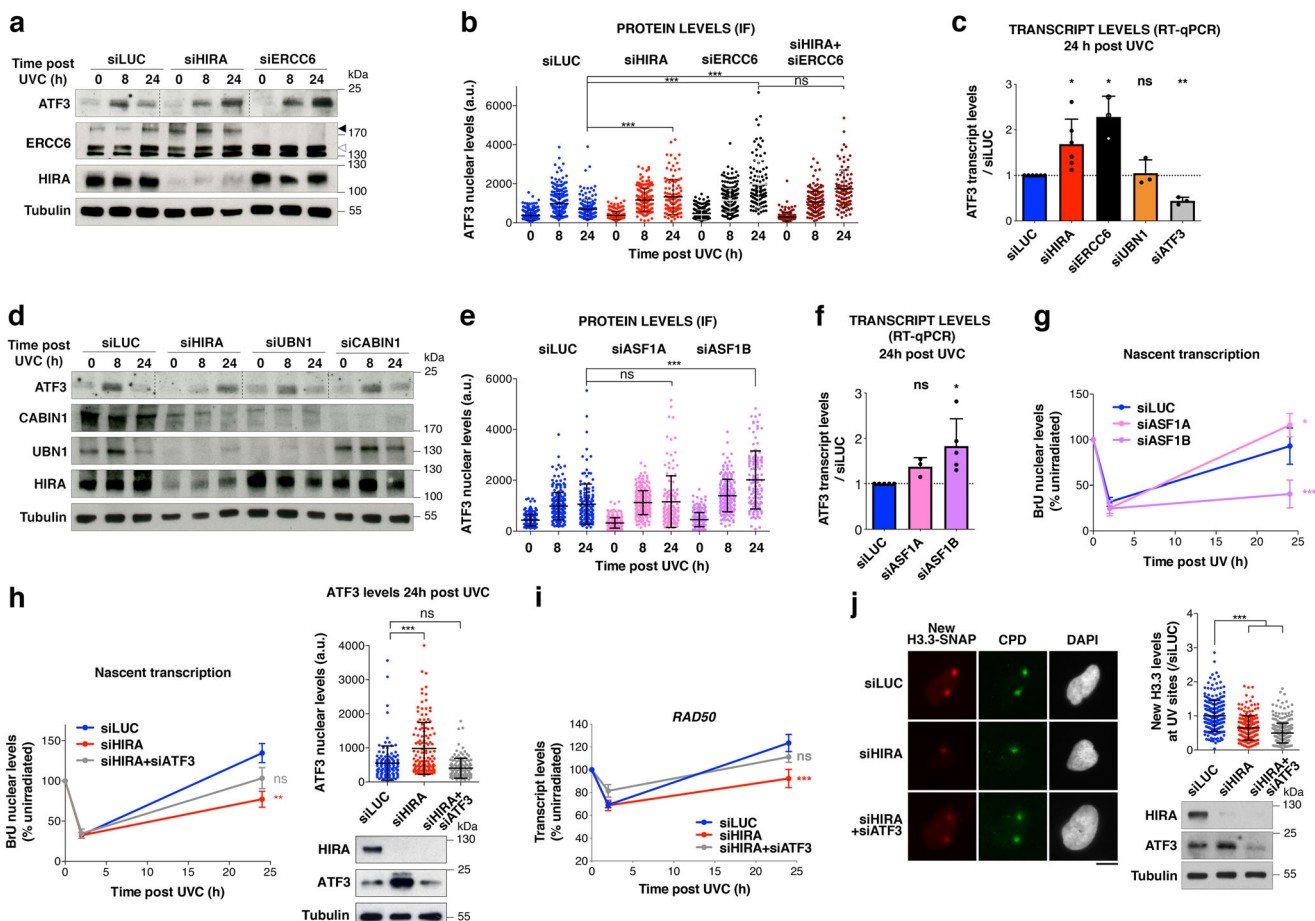

**Fig. 6 HIRA promotes transcription recovery after UVC damage through ATF3 downregulation. a**, **d** Western blot analysis of ATF3 protein levels at the indicated time points post UVC irradiation in HeLa cells treated with the indicated siRNAs (siLUC, control; black arrowhead, ERCC6 full-length; white arrowhead, ERCC6 splice variant; Tubulin, loading control). Dotted lines on the ATF3 panels delineate different siRNA conditions on the same exposure. **b**, **e** ATF3 nuclear levels analysed by immunofluorescence (IF) at the indicated time points post UVC irradiation in HeLa cells treated with the indicated siRNAs (siLUC, control). Knockdown efficiencies are controlled by western blot (Tubulin, loading control; for panel b, refer to Supplementary Fig. 5d). The scatter plots show ATF3 levels in cell nuclei (mean ± s.d. from at least 94 cells). Similar results were obtained in two (**b**) and three (**e**) independent experiments. a.u. arbitrary units. **c**, **f** ATF3 transcript levels (normalised to GAPDH) analysed by RT-qPCR 24 h post UVC in HeLa cells treated with the indicated siRNAs (siLUC, control). Results are presented relative to siLUC (mean ± s.d. from at least three independent experiments). **g**, **h** Nascent transcript levels analysed by bromo-uridine (BrU) incorporation in UVC-irradiated HeLa cells treated with the indicated siRNAs (siLUC, control). The line graphs represent mean values ± s.e.m. from three (**g**) and six (**h**) independent experiments, scoring at least 79 cells per experiment. siRNA efficiencies are controlled by western blot (Tubulin, loading control) and by ATF3 immunofluorescence 24 h post UVC irradiation as shown on the scatter plot (mean ± s.d. from at least 117 cells; similar results were obtained in three independent experiments). **i** Transcript levels analysed by qRT-PCR for the *RAD50* gene product in HeLa cells treated with the indicated siRNA (siLUC, control). Mean ± s.e.m from five independent experiments. **j** New H3.3 accumulation at UVC damage sites (marked by CPD immunostaining) analysed 2 h post local UVC irradiation in U2OS H3.3-SNAP cells treated with the indicated siRNAs (siLUC, control). Knockdown efficiencies are controlled by western blot (Tubulin, loading control). New H3.3 levels at UV sites are shown on the scatter plot (mean ± s.d. from at least 193 cells scored in three independent experiments). a.u. arbitrary units. Statistical significance is calculated by one-way (**b**, **c**, **e**, **f**, **j** scatter plots and bar graphs) and two-way ANOVA (**g**, **h**, **i** line graphs) with Bonferroni posttest *: $p < 0.05$; **: $p < 0.01$; ***: $p < 0.001$; ns: $p > 0.05$. Scale bars, 10 μm. Source data are provided as a Source Data file.

transcripts (Fig. 6h). The transcription rescue was observed for the ATF3 target gene *RAD50* as shown by quantitative RT-PCR (Fig. 6i). Notably, however, ATF3 depletion did not restore new H3.3 deposition at UV sites in HIRA knocked-down cells (Fig. 6j), further strengthening the idea that transcription recovery and new H3.3 deposition are mechanistically and functionally uncoupled. These results demonstrate that HIRA promotes transcription recovery after UVC damage at least in part by controlling the levels of the transcription repressor ATF3.

**HIRA promotes transcription recovery after UVC damage through UBN2 stabilisation independently of the ATF3 pathway.** Considering that the effect of ATF3 knockdown on

alleviating the transcription defect in HIRA-depleted cells was only partial, we assumed that additional, ATF3-independent pathways might mediate HIRA function on transcription recovery post UVC. During the course of our studies, UBN2 was characterised as an alternative subunit to UBN1 within the HIRA complex, chaperoning H3.3 in mouse embryonic stem cells[59]. We confirmed the contribution of UBN2 to new H3.3 deposition at UVC damage sites in human cells (Fig. 7a). Similar to UBN1 depletion (Fig. 1b), UBN2 depletion partially reduced new H3.3 deposition at UV sites and we noticed that combined depletion of UBN1 and UBN2 had a greater effect (Supplementary Fig. 6a), arguing that both subunits have additive rather than redundant roles in new H3.3 deposition. Remarkably, in sharp contrast to

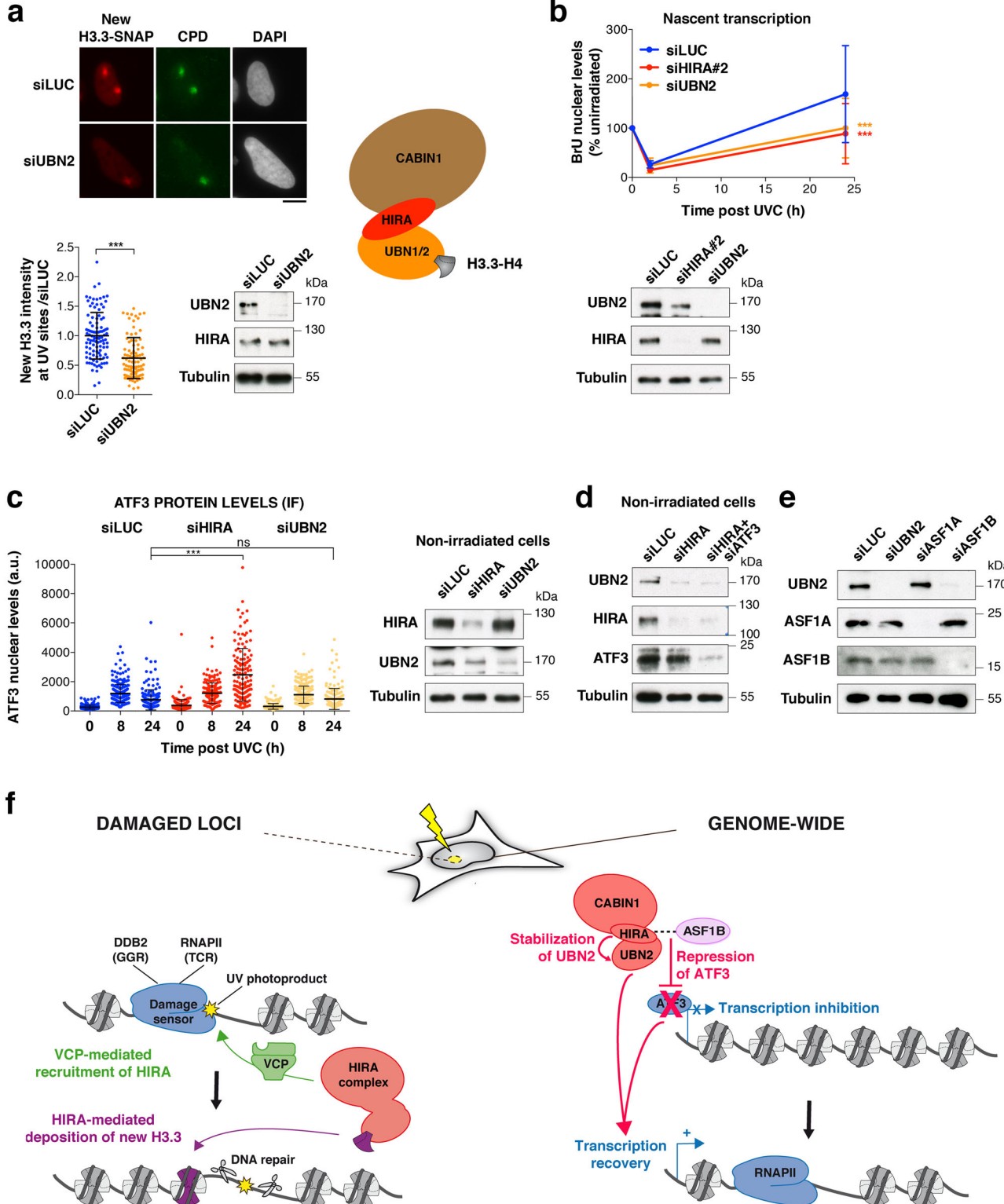

what we observed upon UBN1 depletion, depletion of UBN2 strongly inhibited transcription recovery post UVC, similar to HIRA knockdown and without any detectable impact on cell cycle distribution (Fig. 7b and Supplementary Fig. 1d). We confirmed the defective transcription recovery with a second siRNA against UBN2 (Supplementary Fig. 6b). UBN2 depletion did not affect HIRA levels, while HIRA knock-down led to a down-regulation of UBN2 (Fig. 7b), as observed for the other subunits of the complex (Fig. 1a, b). These findings indicate that

HIRA stabilizes UBN2 and cooperates with UBN2 and not with UBN1 for promoting transcription recovery after UV damage. To test whether UBN2 operated through ATF3 in transcription regulation post UVC, we examined ATF3 levels upon UBN2 depletion. Notably, ATF3 did not display any sustained accumulation 24 h post UVC in UBN2-knocked down cells (Fig. 7c, Supplementary Fig. 6c). Reciprocally, ATF3 knock-down did not rescue UBN2 levels in HIRA-depleted cells (Fig. 7d), thus defining two independent pathways downstream of HIRA for transcription

**Fig. 7 HIRA promotes transcription recovery after UVC damage through UBN2 stabilisation independently of the ATF3 pathway. a** New H3.3 accumulation at UVC damage sites (marked by CPD immunostaining) analysed 2 h post local UVC irradiation in U2OS H3.3-SNAP cells treated with the indicated siRNAs (siLUC, control). siRNA efficiencies are controlled by western blot (Tubulin, loading control). The scatter plots show new H3.3 levels at UV sites (mean ± s.d. from at least 86 cells scored in three independent experiments). The scheme on the right represents the HIRA complex with the alternative H3.3 chaperone subunits UBN1 and UBN2. Scale bar, 10 μm. **b** Nascent transcript levels analysed by bromo-uridine (BrU) incorporation in UVC-irradiated HeLa cells treated with the indicated siRNAs (siLUC, control). Knockdown efficiencies are controlled by western blot (Tubulin, loading control). The graph shows mean values ± s.d. from three independent experiments scoring at least 81 cells per experiment. **c** ATF3 nuclear levels analysed by immunofluorescence (IF) at the indicated time points post UVC irradiation in HeLa cells treated with the indicated siRNAs (siLUC, control). The scatter plot shows mean values ± s.d. from at least 94 cells. Similar results were obtained in three independent experiments. a.u. arbitrary units. Statistical significance is calculated by two-sided Student's $t$-test (**a**), two-way (**b**) and one-way ANOVA (**c**) with Bonferroni posttest. ***: $p < 0.001$; ns: $p > 0.05$. **d–e** Western blot analysis of UBN2 levels in HeLa cells treated with the indicated siRNAs (Tubulin, loading control). Similar results were obtained in three independent experiments. **f** Model for HIRA function in transcription recovery post UVC damage. Early after UVC damage, the HIRA complex (red) is recruited by VCP (green) to UV-damaged chromatin where it mediates the deposition of new H3.3 histones (purple). This local action of the HIRA complex at UV-damaged loci does not contribute to transcription restart, which most likely relies on the release of UV-stalled RNAPII. The HIRA subunit controls a later and broader transcriptional response to UV damage, which is not restricted to genes harbouring UV lesions. HIRA coordinates with ASF1B to stabilise UBN2 and to reduce the expression of the transcription repressor ATF3, both pathways independently contributing to transcription restart. Source data are provided as a Source Data file.

recovery following UVC damage. Given that HIRA coordinates with ASF1B to control ATF3 expression, we also tested the impact of ASF1B on UBN2 levels. We observed that knockdown of ASF1B and not that of ASF1A resulted in reduced UBN2 protein levels (Fig. 7e). This indicates that HIRA cooperates with ASF1B to maintain UBN2 levels. Collectively, these data identify the UBN2 subunit of the HIRA complex as critical for transcription restart after UVC damage, independently of ATF3.

## Discussion

Transcription is tightly regulated in response to DNA damage, which restricts the production of aberrant transcripts from damaged loci and prevents deleterious collisions between transcription and repair machineries[3,11,86]. Despite several decades of research, how transcription restarts after DNA damage repair in a chromatin context is not fully elucidated[8]. Our work makes several important contributions to our mechanistic understanding of this process in human cells and also redefines the role of the histone chaperone complex subunit HIRA. (1) First, we uncover a non-canonical function for HIRA in transcription regulation following UVC damage, which does not involve new H3.3 deposition. (2) Furthermore, we show that HIRA primes UV-damaged chromatin for transcription restart by relieving transcription inhibition rather than by depositing an activating bookmark. HIRA indeed operates, at least in part, through reducing the levels of the transcription inhibitor ATF3. (3) Thus, we delineate two functionally distinct HIRA-dependent pathways: a local and early response to UV damage resulting in H3.3 deposition, and a global and late response leading to transcription restart on a genome-wide scale (Fig. 7f).

**VCP-dependent recruitment of HIRA to damaged loci**. We have identified the ubiquitin-binding factor VCP as a driver of HIRA recruitment to UV-damaged loci, which provides a molecular basis for this ubiquitin-dependent process[29]. How VCP mediates HIRA recruitment to UV-damaged chromatin is not fully elucidated. This may involve a physical interaction between the two proteins as observed for the fission yeast orthologs[87] or an indirect mechanism relying on the ability of HIRA to bind naked DNA[44] that may be exposed upon VCP-mediated eviction of ubiquitylated proteins from damaged sites. Noteworthy, recent findings revealed a trimerisation of HIRA subunit, critical for HIRA recruitment to UVC-damaged chromatin and for new H3.3 deposition[88]. How HIRA trimerisation crosstalks with VCP to mediate HIRA recruitment to damaged chromatin could be tested in future studies.

**Transcription-independent function of HIRA at damaged loci**. An important conclusion from our work is the discovery that HIRA promotes transcription restart after UVC damage independently of new H3.3 histone deposition at UV damage sites. HIRA-mediated deposition of H3.3 is also dispensable for GGR of UV lesions[29]. Thus, the functional relevance of new H3.3 deposition by HIRA in UV-damaged chromatin is still elusive. It might contribute to restrict pervasive transcription[89] upon DNA damage or play a transcription-independent role that is still to be determined. Recently, a novel function for the HIRA complex in recycling parental H3.3 histones during transcription was reported in human cells[90], which shares interesting similarities with the function of HIRA that we describe here. Indeed, HIRA was shown to recycle parental H3.3 in an ASF1-dependent and UBN1-independent manner[90], raising the possibility that parental H3.3 recycling rather than new H3.3 deposition could be involved in HIRA-mediated transcription restart post UV. Not only is HIRA-mediated deposition of new H3.3 at UV-damaged loci dispensable for transcription recovery, but HIRA recruitment to UV-damaged chromatin is also transcription-independent[29], arguing that HIRA function at UV-damaged sites is not connected to transcription regulation. It is reasonable to assume that at damaged loci the release of UV-stalled elongating RNAPII largely contributes to transcription recovery. Indeed, transcription restart after UVC damage in human cells was shown to operate in a 5′ to 3′ wave from the transcription start site of genes[68] rather than from sites of RNAPII stalling, highlighting the need for a release of UV-stalled RNAPII to allow transcription restart/re-initiation. In line with this, genome-wide analyses suggest that RNAPII is released from its template during TCR in human cells[18]. In addition to RNAPII release, ubiquitin-dependent proteolysis of UV-stalled RNAPII regulates the pool of available RNAPII for transcription initiation and plays a pivotal role in transcription regulation post UV in mammalian cells[21,22]. RNAPII degradation indeed promotes both the silencing of short genes and transcription restart[21,22], showing intriguing similarities with HIRA impact on transcription post UV. However, we did not notice any defect in the degradation of UV-stalled RNAPII upon HIRA loss-of-function and previous work showed that RNAPII levels returned to normal 24 h post UVC in the absence of HIRA[29], indicating that impaired repression of short genes and defective transcription recovery in this context are not attributable to perturbations in the pool of available RNAPII.

**Genome-wide impact of HIRA on transcription restart post UV**. Distinct from the local action of HIRA at UV damage sites,

our findings reveal a broad effect of HIRA on transcription recovery, including that of very short genes, which are less likely to bear UV lesions. Note that we cannot rule out the possibility that some regulatory elements lying outside these short genes are damaged. Nevertheless, these data suggest that the function of HIRA is not restricted to the vicinity of damage sites (cis-effect) and reflect a more global impact of HIRA on nuclear transcription (trans-effect). In line with this, we found that the CABIN1 subunit of the HIRA complex, which stimulates HIRA recruitment to UV damage sites[29], is dispensable for transcription recovery following UVC irradiation. This further strengthens the idea that HIRA function in transcription restart is mechanistically distinct from its recruitment to UV damaged loci. Similar to what was reported for the TCR factor ERCC6[31,81], we have shown that the global function of HIRA in transcription recovery post UVC damage is at least partly mediated by the transcription inhibitor ATF3, whose levels are reduced at late time points after damage in a HIRA-dependent manner. Despite the similarities between HIRA and ERCC6, we demonstrate that HIRA is not a bona fide TCR component. In line with this, HIRA knocked-down cells do not display increased sensitivity to UVC unlike what is observed upon loss-of-function of TCR factors[29]. We found that HIRA coordinates with ASF1B to silence ATF3, potentially by recruiting transcriptional repressors to the *ATF3* gene. This result is intriguing considering that HIRA preferentially associates with ASF1A and not ASF1B in unchallenged cells[43,84,85], thus highlighting a specific functional interaction between HIRA and ASF1B during the UV-damage response. In line with this, we observed increased levels of ASF1B and not ASF1A after UV damage. Noteworthy, ASF1B did not show any detectable accumulation at UV damage sites[29], further strengthening the idea that the HIRA-ASF1B-ATF3 axis regulates transcription recovery on a genome-wide scale rather than at damaged loci. Supporting a contribution of the ATF3 pathway to HIRA-dependent transcription recovery post UVC, we have shown that ATF3 depletion rescues transcription recovery in the absence of HIRA. The restoration is only partial, likely due to the fact that not all UV-repressed genes are ATF3 targets. Furthermore, UBN2 also contributes to HIRA transcriptional function independently of the ATF3 axis. Our data thus delineate two independent pathways, HIRA-ASF1B-ATF3 and HIRA-ASF1B-UBN2, which both contribute to transcription recovery post UV damage. Regarding UBN2, in addition to binding and depositing H3.3 at regulatory elements of developmental genes in mouse embryonic stem cells[59], we now show that UBN2 also promotes de novo deposition of H3.3 at UV damage sites in human cells. Nevertheless, this histone deposition activity is not involved in transcription recovery post UV damage, arguing that transcription restart relies on another function of UBN2, yet to be identified. So far, little is known about UBN2, impeding full characterisation of the underlying mechanisms. Considering the genome-wide effect of HIRA on transcription regulation post UVC, it is tempting to speculate that HIRA may also stimulate transcription induction of early UV-responsive genes. Lending support to this idea, the ortholog of HIRA in fission yeast is required for the induction of stress-responsive genes[91]. Future studies will address whether this function of HIRA is conserved in higher eukaryotes.

**A new perspective on the HIRA histone chaperone complex.** Our analysis of HIRA complex subunits showed that only HIRA and UBN2, and not UBN1 and CABIN1, are needed for transcription recovery post UVC. These results are in line with a multiomic analysis of the UV response in human fibroblasts, which identified HIRA and UBN2 but not UBN1 and CABIN1 as required for transcriptional activity following UVC damage[41].

These findings highlight important functional differences between the related subunits UBN1 and UBN2, which deserve further investigation. Whether the HIRA subunit functions in transcription restart in complex with CABIN1, UBN1/2 and H3.3 or as part of another protein complex is still to be determined. Recent biochemical and genomic data suggest that HIRA may operate independently of the histone chaperone complex in some instances. Indeed, gel filtration analyses of the HIRA complex in human embryonic stem cells identified a fraction containing the HIRA subunit but not CABIN1 nor H3.3[92]. In addition, ChIP-seq analyses revealed that HIRA binds more than a thousand sites across the human genome in the absence of UBN1 and these sites are not enriched in H3.3[83], pointing to a function of HIRA independent of the H3.3-chaperone complex, which deserves additional investigation. Further distinguishing the HIRA subunit from CABIN1 and UBN1, mutations in the plant ortholog of HIRA lead to a significantly more pronounced developmental phenotype than mutations in UBN1 or CABIN1[93]. Interestingly, like HIRA, other histone chaperones also control responses to DNA damage independently of their histone binding activity, as shown for ASF1A in DNA double-strand break signalling[94]. Histone chaperones thus appear as multifunctional proteins, which are pivotal for coordinating chromatin dynamics with critical nuclear functions such as transcription and damage signalling.

## Methods

**Cell culture and drug treatments.** HeLa (American Type Culture Collection ATCC CCL-2, human cervical carcinoma, female), XP4PA-SV (Coriell Institute for Medical Research, GM15983, human XPC-deficient skin fibroblasts, male), and U2OS cells (ATCC HTB-96, human osteosarcoma, female) were grown at 37 °C and 5% $CO_2$ in Dulbecco's modified Eagle's medium (DMEM, Life Technologies) supplemented with 10% foetal bovine serum (EUROBIO), 100 U/ml penicillin and 100 µg/ml streptomycin (Life Technologies). U2OS cells stably expressing H3.3-SNAP[95] were maintained in the same medium in the presence of 100 µg/ml G418 (Invitrogen).

The VCP inhibitor NMS-873 (5 µM final concentration, Selleckchem) and the transcription inhibitor Flavopiridol (10 µM final concentration, Sigma-Aldrich) were added to the culture medium 2 h before UV irradiation of the cells and kept until the time of cell fixation.

**siRNA sequences.** siRNA purchased from Eurofins MWG Operon (Supplementary Table 1) were transfected into cells using Lipofectamine RNAiMAX (Invitrogen) following the manufacturer's instructions. The final concentration of siRNA in the culture medium was 50 nM. Cells were harvested 48–72 h post transfection (24–48 h for siH3.3).

**SNAP labelling of histones.** For specific labelling of newly synthesised histones, pre-existing SNAP-tagged histones were first quenched by incubating cells with 10 µM of the non-fluorescent SNAP reagent (SNAP-cell Block, New England Biolabs) for 30 min at 37 °C followed by a 30 min wash in fresh medium and a 2 h chase. Cells were then locally irradiated with the UVC lamp and the SNAP-tagged histones neosynthesized during the chase time were pulse-labelled by incubating cells with a fluorescent SNAP reagent (2 µM SNAP-cell TMR star or 4 µM SNAP-cell Oregon green, New England Biolabs) for 15 min at 37 °C, followed by 45 min–1 h 45 min incubation in fresh medium. Cells were then processed for immunostaining. For cells harvested 24 h post irradiation, the quenching reagent was added again after the pulse step to avoid labelling of additional histones by free fluorescent SNAP reagent remaining in the cells.

**UV irradiation.** Cells were irradiated with UVC (254 nm) using a low-pressure mercury lamp (Vilbert-Lourmat). Conditions were set using a VLX-3W dosimeter (Vilbert-Lourmat). For global irradiation, cells in PBS (Phosphate Buffer Saline) were exposed to 10 J/m², except for analysing RNAPII degradation (Fig. 3) where the dose was 50 J/m². For local irradiation[10], cells on glass coverslips (VWR) were covered with a polycarbonate filter (5 µm pore size, Millipore) and irradiated with 150 J/m² UVC for analysing HIRA and H3.3 accumulation at UV sites or with 500 J/m² UVC in the TCR assay and when staining for ATF3 and VCP.

**Cell extracts and western blot.** Total extracts were obtained by scraping cells on plates or resuspending cell pellets in Laemmli buffer (50 mM Tris-HCl pH 6.8, 1.6% Sodium Dodecyl Sulfate, 8% glycerol, 4% β-mercaptoethanol, 0.0025% bromophenol blue) followed by 5 min denaturation at 95 °C.

For western blot analysis, extracts were run on 4–20% Mini-PROTEAN TGX gels (Bio-Rad) in running buffer (200 mM glycine, 25 mM Tris, 0.1% SDS) and transferred onto nitrocellulose membranes (Protran) with a Trans-Blot SD semi-dry transfer cell (Bio-Rad). Total proteins were revealed by Pierce® Reversible Stain (Thermo Scientific). Proteins of interest were probed using the appropriate primary and HRP (Horse Radish Peroxidase)-conjugated secondary antibodies (Supplementary Table 2), detected using SuperSignal West Pico or Femto chemiluminescence substrates (Pierce) on hyperfilms MP (Amersham) or with Odyssey Fc-imager (LI-COR Biosciences). Alternatively, when fluorescence detection was used instead of chemi-luminescence, total proteins were revealed with REVERT total protein stain, secondary antibodies were conjugated to IRDye 680RD or 800CW (Supplementary Table 2) and imaging was performed with Odyssey Fc-imager (LI-COR Biosciences). Levels of proteins of interest were then quantified with Image Studio Lite software using total protein stain for normalisation.

**Immunofluorescence**. Cells grown on glass coverslips (VWR) were either fixed directly with 2% paraformaldehyde and permeabilised with 0.2% Triton X-100 in PBS, or pre-extracted before fixation with 0.5% Triton X-100 in CSK buffer (Cytoskeletal buffer: 10 mM PIPES pH 7.0, 100 mM NaCl, 300 mM sucrose, 3 mM MgCl2). Pre-extraction was required only for visualising new H3.3-SNAP deposition at damage sites. For CPD staining, DNA was denatured with 0.5 M NaOH for 5 min. Samples were blocked in 5% BSA (Bovine Serum Albumin, Sigma-Aldrich) in PBS supplemented with 0.1% Tween before incubation with primary and secondary antibodies conjugated to Alexa–Fluor dyes (Molecular Probes) (Supplementary Table 2). Coverslips were mounted in Vectashield medium with DAPI (Vector laboratories).

**Image acquisition and analysis**. Epifluorescence imaging was performed with a Leica DMI6000 epifluorescence microscope using a Plan-Apochromat 40x/1.4 oil or 63x/1.3 oil objective. Images were captured using a CCD camera (Photometrics) and Metamorph software. Images were analysed with ImageJ (National Institutes of Health, Bethesda, Maryland, USA, http://imagej.nih.gov/ij/) and ICY (Institut Pasteur, Paris, France, http://icy.bioimageanalysis.org/) softwares.

**Flow cytometry**. For cell cycle analysis, cells were fixed in ice-cold 70% ethanol before DNA staining with 50 µg/ml propidium iodide (Sigma-Aldrich) in PBS containing 0.05% Tween and 0.5 mg/ml RNase A (USB/Affymetrix). The DNA content was analysed by flow cytometry using a BD FACScalibur flow cytometer and CellQuest Pro software (BD Biosciences). The cell cycle distribution was estimated with the Watson model on FlowJo software (TreeStar).

**Transcription-coupled repair (TCR) assay**. GGR-deficient XP4PA-SV cells were starved overnight in medium containing 0.5% serum to reduce the number of replicating cells. The following day, the cells were subject to local UVC irradiation at 500 J/m$^2$ and incubated with 20 µM 5-ethynyl-2'-deoxyuridine (EdU, Thermo-Fisher Scientific) and 1 µM 5-Fluoro-2'-deoxyuridine (FUdR, Sigma-Aldrich) for 3.5 h at 37 °C. FUdR decreases endogenous thymidine production thus favoring EdU incorporation. EdU incorporation was revealed with the Click-iT EdU Imaging kit (ThermoFisher Scientific) using Alexa Fluor 594 dye according to the manufacturer's instructions. After EdU revelation, cells were fixed and processed for CPD detection by immunofluorescence. Quantification of repair synthesis was performed on Image J software using a custom macro: S phase cells were excluded from the analysis based on pan-nuclear EdU signal and then EdU and CPD intensities were measured at the damage sites (delineated based on CPD signal, with manual adjustment of the detection threshold).

**Nascent RNA imaging**. Cells were exposed to global UVC irradiation (10 J/m$^2$) at different time points post siRNA transfection (48 h or 70 h) to be subject to nascent RNA labelling all at the same time (71 h post siRNA). siRNA efficiency was confirmed both at 48 h (time of the first irradiation) and 72 h post-transfection (time of fixation). Cells were incubated in DMEM supplemented with 0.5 mM ethynyl-uridine (EU) for 1 h at 37 °C, rinsed in PBS before fixation in 2% paraformaldehyde. EU incorporation was revealed with Click-iT RNA Imaging kit (Thermo-Fisher Scientific) using Alexa Fluor 594 dye according to the manufacturer's instructions. Alternatively, cells were incubated with 4 mM bromouridine (BrU) in conditioned DMEM for 30 min at 37 °C, followed by detergent extraction with 0.5% Triton X-100 in CSK buffer and fixation in 2% paraformaldehyde. BrU was revealed by immunofluorescence using an anti-BrdU antibody (Supplementary Table 2). Coverslips were mounted in Vectashield medium with DAPI (Vector laboratories). The mean fluorescence intensity per nucleus was obtained using ImageJ software. Nuclear segmentation was based on DAPI staining.

**Nascent RNA Bru-seq**. For nascent RNA sequencing from UV-irradiated cells[68], HeLa cells exposed to 10 J/m$^2$ UVC were allowed to recover 0, 2, 6, 24 h before incubation with 2 mM bromouridine (BrU) for 30 min at 37 °C. Cells were exposed to global UVC irradiation at different time points post siRNA transfection (48 h, 66 h, 70 h or 72 h) to be subject to nascent RNA labelling with BrU all at the same

time (72 h post siRNA). Non-irradiated cells were used as controls. The cells were lysed in TRIzol reagent (Invitrogen) and BrU-containing RNA was isolated on goat anti-mouse IgG Dynabeads (Invitrogen, 50 µl/condition) coupled with 2 µg anti-BrdU antibody[67] (Supplementary Table 2). cDNA libraries were made from the BrU-labelled RNA using the Illumina TruSeq library kit and sequenced using Illumina HiSeq sequencers at the University of Michigan DNA Sequencing Core with 52 bp single-end reads (on average 50 million reads per sample with 70% uniquely mapping reads). Sequencing and read mapping (uniquely mapping reads only) were carried out using the human genome reference assembly hg19/GRCh37 with Illumina Casava v.1.8.2 for base calling, Bowtie v.0.12.8 and TopHat v.1.4.1 for read mapping, Bedtools v.2.16.2 for genome annotation[67]. All primary sequencing data files have been deposited in NCBI's Gene Expression Omnibus (GEO) with the accession number GSE151833.

We employed biological scaling normalisation[69] because RNA sequencing data are normalised to total read counts in each sample (RPKM), which is not suited for the detection of global expression changes such as those observed after UVC irradiation. For this, we multiplied all RPKM values by correction factors corresponding to mean nascent RNA amounts per cell nucleus measured by EU staining for each siRNA condition and at each time point post UVC (table in Supplementary Fig. 3b). Only genes with a detectable transcription level in non-irradiated cells (RPKM > 0) were analysed. After biological scaling normalisation, we considered as UV-repressed, genes with at least a 2-fold reduction in nascent transcripts 2 h 30 min after UVC irradiation compared to non-irradiated cells in control conditions (siLUC) and as UV-induced, those with at least 2-fold induction in nascent transcripts 30 min after UVC irradiation.

To determine the dependency of UV-repressed genes on HIRA or ERCC6 for transcription recovery, we defined as independent genes those whose transcription recovery 24h30 post UVC in siHIRA/ERCC6 was equal to the transcription recovery observed in siLUC +/-10%. Stimulated genes were those with at most 90% transcription recovery in siHIRA/ERCC6 compared to siLUC, and inhibited genes were those with at least 110% better recovery in siHIRA/ERCC6 compared to siLUC. Gene ontology analyses were performed with GOrilla[96] (database version Jan 19, 2019) on UV-repressed genes that recover transcription better in the absence of HIRA compared to all UV-repressed genes as background (only common genes from experiments #1 and #2 were included). The positions of ATF3 ChIP-seq peaks (with FDR < 1, fold enrichment >10)[31] were intersected with the positions of the nascent transcripts extended by 5 kb upstream to include promoter regions.

**ChIP-seq data representation**. ChIP-seq data for HIRA binding to the *ATF3* gene in HeLa cells were from Pchelintsev et al., 2013[83]. Log2 fold change relative to input was obtained by bigwigCompare and plotted with Integrative Genomics Viewer (IGV 2.3). Regulatory regions were defined by enrichments in H3K4me1 and H3K27ac marks based on ENCODE data.

**Quantitative RT-PCR**. For qRT-PCR on total RNA, RNA was extracted from cells with TRIzol™ Reagent following the manufacturer's instructions (Invitrogen) and precipitated in isopropanol. RNA samples were subject to DNA digestion with Turbo DNA-free (Invitrogen) before reverse transcription with Superscript III RT using random primers (200 ng/reaction, Invitrogen). Quantitative PCR reactions were carried out with the indicated exonic primer pairs (Eurofins MWG Operon, 500 nM final concentration, Supplementary Table 3) and the Power SYBR® Green PCR Master Mix (Applied Biosystems) and read in MicroAmp® Fast Optical 96-well plates (Applied Biosystems) using a ABI 7500 Fast detection system (Applied Biosystems). Results were normalised to the amount of the *GAPDH* housekeeping gene product, a long half-life transcript, whose levels remain fairly stable over 24 h post UVC irradiation.

For qRT-PCR on BrU-labelled RNA, cells were incubated with 4 mM bromouridine (BrU) for 30 min at 37 °C before RNA extraction as above. Turbo reaction products were used for pull-down of BrU-labelled RNA using goat anti-mouse IgG Dynabeads (Invitrogen, 25 µl/condition) coupled with 1 µg anti-BrdU antibody[67] (Supplementary Table 2). The specificity of the pull-down reaction was confirmed by including a sample of cells that were not incubated with BrU as a negative control. Quantitative PCR reactions were carried out as above on pulled-down material. Input material taken from Turbo reactions (5%) was used for GAPDH amplification and normalisation.

**Statistical analyses**. Heatmaps were established with R software. Overlaps on Venn diagrams were analysed by Fisher's exact test using R software. Other statistical tests were performed using GraphPad Prism. $p$ values for mean comparisons between two groups were calculated with a two-sided Student's $t$-test with Welch's correction. Multiple comparisons were performed by one-way ANOVA with Bonferroni posttest. ATF3 transcript levels relative to siLUC were compared to a theoretical mean of 1 by one-sample $t$-test (two-sided). Analyses of transcription recovery 24 h post UVC (EU or BrU incorporation and RT-qPCR assays) were performed by two-way ANOVA with Bonferroni posttest. Correlations were evaluated using the Spearman rank test. ns: non-significant, *: $p < 0.05$, **: $p < 0.01$, ***: $p < 0.001$. Confidence interval of all statistical tests: 95%.

**Reporting summary**. Further information on research design is available in the Nature Research Reporting Summary linked to this article.

## Data availability

All data generated during this study are included in this article and its supplementary information files. The accession number for primary sequencing data files is GSE151833. Source data are provided with this paper and are available on Mendeley https://doi.org/10.17632/njd29jsdfr.1. Source data are provided with this paper.

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

## Acknowledgements
We thank members of our laboratory for stimulating discussions and Claire Francastel, Valérie Mezger and Claire Rougeulle for critical reading of the manuscript. We thank Artur Veloso, Brian Magnuson and Magali Hennion for helping with bioinformatics analyses. We acknowledge the imaging platform of the Epigenetics and Cell Fate Centre. This work was supported by the European Research Council (ERC grants ERC-2013-StG-336427 "EpIn" and ERC-2018-CoG-818625 "REMIND"), the French National Research Agency (ANR-12-JSV6-0002-01), the "Who am I?" laboratory of excellence (ANR-11-LABX-0071) funded by the French Government through its "Investments for the Future" program (ANR-11-IDEX-0005-01), EDF Radiobiology program RB 2014-01, the Fondation ARC. S.E.P. is an EMBO Young Investigator. J.F. is recipient of PhD fellowships from University Paris Diderot and EUR G.E.N.E (ANR-17-EURE-0013).

## Author contributions

D.B., J.F. and S.E.P designed and performed experiments with technical assistance from O.C., and analysed the data. M.P., M.L. and S.E.P performed Bru-seq experiments and bioinformatic analyses. S.E.P. supervised the project and wrote the manuscript with inputs from all authors.

## Competing interests
The authors declare no competing interests.
