## [Peer Review File · Nature Communications]

REVIEWER COMMENTS

Reviewer #1 (Remarks to the Author):

This manuscript reports a requirement for HIRA, as well as ASF1B and UBN2, in transcription restart after UV damage repair, but is independent of UBN1 or ASF1A. They also report that loss of HIRA or ASF1B leads to elevated levels of the transcriptional repressor ATF3 which partly accounts for the failure to restart transcription after UV repair. In general the experiments are well designed and the story is interesting, although there are a few inconsistencies and gaps that need filling to make a coherent story. They have not convincingly shown that this is multiple pathways, nor involves a non-canonical pathway, as opposed to one canonical pathway for HIRA.

Major:

- All of the transcription data throughout the paper are normalized to 100% for each cell line in the absence of DNA damage. It is possible that depletion of HIRA per se leads to a global increase in transcription regardless of DNA damage. It is important that they show at least some experiments without normalization to 100% so the global effects on transcription before UV damage can be assessed, as it will affect the interpretation of all the data.
- It is concluded that new H3.3 is not required for transcriptional restart after UV damage, however this is not convincing because the depletion of H3.3 is not complete in Fig. 1D. This caveat needs to be mentioned in the text, because the remaining H3.3 may be sufficient for transcriptional restart. Similarly, in the experiment in Fig. D, the extent of transcriptional repression upon UV damage is a lot less than in other experiments, going down to about 65% vs 40% in all the other experiments. This difference seems to be attempted to be hidden by the altered scale on the vertical axis. All the EU fluorescence graphs should be shown with an axis from 0 to 100% so as not to infer a greater effect, which is what happens when one starts the axis at 20% or 40%, as they do in this study.
- They conclude that loss of HIRA leads to elevated levels of the ATF3 transcriptional repressor. That is shown in Fig. 6A and D and H at 24 hours, but the levels of ATF3 are actually reduced in westerns Fig. 6G and Fig. 7D upon HIRA depletion. Consistent results need to be shown.
- In Fig. 6A and D, there are much more increase of ATF3 protein levels at 24 hours than 8 hours in siHIRA cells. However, nascent transcription level in siHIRA cells at 24 hours is higher than that at 8 hours in Fig. 6G and all other figures, which is not consistent with the idea that ATF3 represses transcription after damage.
- What do they speculate to be the molecular mechanism whereby loss of ASF1B or HIRA leads to elevated ATF3 transcript levels 24 hours after UV damage?
- Given that HIRA depletion leads to loss of UBN2 protein, the results reported for HIRA could be indirect effects that are a consequence of loss of UBN2. This could be tested by seeing if reexpression of UBN2 is sufficient to restore transcription restart in a HIRA depleted cell.
- Does loss of UBN2 also lead to elevated ATF3 protein levels?
- Similarly they show that HIRA depletion does not affect ASF1A levels, but they need to show whether this is the case with ASF1B or not.
- It is a bit of a stretch to say that these are non canonical roles for HIRA, given that they did not rule out a role for parental H3.3 incorporation, only new H3.3. Nor did they rule out a role for UBN2 or ASF1B in ATF3 expression, which if they were involved would make it a canonical role for HIRA. As such all references to a non canonical roles for HIRA should be removed.
- The model figure shows new H3.3 promoting transcription restart but that is the opposite of what is concluded in the paper.
- Their model shows that ASF1B and HIRA and UBN2 function together to promote transcription restart. However, they have not shown a requirement for ASF1B in transcription restart after UV damage. It is critical that they examine whether ASF1B is required for transcriptional restart or not after UV damage.
- The data shown with the EU incorporation measured by fluorescence throughout the paper in the absence of HIRA clearly show a profound defect in transcriptional restart. However, that is not apparent in the BrU seq analyses in Fig. 4D and Fig. 5D, where the defect without HIRA appears quite subtle compared to siLUC. The authors need to explain the discrepancy.
- It is possible that all the factors shown function in the same pathway to promote transcription restart after UV, as this has not been addressed. This seems particularly likely given that depletion of UBN2 is equally defective as depletion of HIRA for transcriptional restart. Does depletion of ATF3

also partially reverse the transcription restart defect seen upon UBN2 depletion?

Minor:

- Fig. S4d mRNA levels can not reflect nascent RNA production even of short half-life transcripts. So, the authors should use intron-exon junction primers to validate the results from BrU-seq.
- Fig. 5c, the authors claim that "HIRA-knock-down not only impairs transcription recovery but also mitigates transcription repression of short genes", which might not be true. Because there are much higher basal transcription levels in siHIRA cells compared to controls at earlier time points.

Reviewer #2 (Remarks to the Author):

The manuscript by Boivier et al. from the Polo lab dissects regulatory pathways by which the histone chaperone HIRA regulates transcription recovery following DNA damage repair. This is a thoughtful and elegant follow-up of previous work from the Almouzni lab showing that HIRA is involved in transcription recovery, and implying that this involved new histone H3.3 deposition acting as an early bookmarking mechanisms to restart transcription later on (Adam et al. Cell. 2013).

The current study shows that the deposition of H3.3 is not involved in transcription recovery. Rather HIRA regulates transcription restart by two distinct mechanisms: (1) shutting down ATF3 transcription at late time-points after UV (24 hrs), to relieve its repressive impact and enable cells to re-initiate transcription, and (2) stabilize the protein levels of regulatory subunit UBN2. The study provides important insights into how cells recover transcription after DNA damage repair is completed by revealing that HIRA regulates ATF3 expression at the transcriptional level. This reveals an additional regulatory layer by showing that DNA repair itself is not sufficient to restart transcription.

The study does not provide mechanistic insight into how the HIRA/ATF3-dependent pathway works, and in particular how the HIRA/UBN2/ATF3-independent pathway operates. The regulation of ATF3 expression in response to UV damage is complex and regulated at multiple levels. Fully understanding his regulatory network will require more follow-up work. However, this study is a first important step to better understand the mechanisms involved.

The manuscript is logical and well-written, the figures are very clear, and the experiments are thoughtful. This work deserves being considered in Nature Communications, should the authors appropriately address the following points:

Major Points

1. The recruitment of HIRA to sites of local UV damage was previously shown to depend on global genome repair (GGR) factor DDB2 (Adam et al. Cell. 2013; Fig S5A). This links HIRA recruitment to GGR rather than to TCR. The role of VCP in recruiting HIRA would also fit with this scenario, considering that VCP is recruited by and involved in handling ubiquitylated DDB2 at sites of local damage (Puumalainen et al. Nat Comms. 2014, citation 63 in this study). I am not convinced that the recruitment of HIRA is linked to VCP-dependent extraction of RNAPII as suggested by the model depicted in Fig 7f. After all, HIRA recruitment to sites of local UV damage is independent of transcription (Adam et al. Cell. 2013). The authors should address whether H3.3 deposition is dependent on DDB2 (using siRNAs against DDB2 as done in Adam et al. Cell. 2013 (Fig S5A)). It seems as if the authors are deliberately vague on this issue ("HIRA is recruited to damaged chromatin regions in a ubiquitin-dependent manner"). If DDB2 regulates H3.3 deposition, it would strengthen the paper to functionally link HIRA and H3.3 function to GGR, for instance by a UDS assay in wild-type cells (including DDB2 as a control), and revise the model in Fig 7f accordingly. It is possible that the local role of HIRA stimulates GGR, while it has a global impact downstream from TCR in regulating transcription recovery.

2. I applaud the use of the biological scaling normalization (in this case RRS experiments) to normalize the BrU-seq data. The validation by qRT-PCR of several genes (Fig S4d) makes a strong case that this approach is valid and matches the normalized BrU-seq data very well. I suggest to

show the BrU-seq and qRT-PCR data for one gene next to each other in the main figure. The BrU-seq data also contains information on the distribution of nascent transcripts within the genome, which is not really used in this study. It could be informative to show the BrU-seq read distribution from siLUC and siHIRA cells mapped on the genome for at least one gene at different time-points after UV as an example (e.g. RAD50 or NIPBL).

3. The authors argue that ASF1B cooperates with HIRA in both pathways (HIRA-ASF1B-ATF3 and HIRA-ASF1B-UBN2). They show in Fig 6e that knockdown of ASF1B affects ATF3 proteins levels, while ASF1A knockdown does not. The authors should test if ASF1B knockdown leads to defective transcription recovery, while siASF1A does not.

4. The partial restoration of transcription recovery measured by BrU incorporation in Fig 6g in siHIRA+siATF3 cells compared to siHIRA cells is very elegant. However, in this experiment there appears to be a partial recovery in siHIRA cells, which is not observed in other experiments (Fig 1a, Fig 2d). Is this also seen under optimized conditions that show a stronger defect of siHIRA? The authors should also monitor the expression of an ATF3 target gene by qRT-PCR (e.g. NIPBL or RAD50) under siHIRA versus siHIRA+siATF3 conditions to see whether a (partial) rescue is observed. This would considerably strengthen the RRS data in Fig 6g.

5. The authors should speculate in the discussion how they think HIRA might regulate ATF3 transcription after UV? Fig S5e shows published CHIP-seq data showing possible HIRA binding to the ATF3 gene promoter and nearby regulatory elements (in unirradiated cells). However, ATF3 expression is not increased in unirradiated cells depleted of HIRA (Fig 6b), meaning that loss of HIRA only leads to elevated ATF3 levels (transcript and protein) after UV irradiation. The authors are invited to include ideas / speculation on how this could work in the discussion.

Minor Points

6. There is considerable variation in the Recovery of RNA synthesis (RRS) experiments when comparing Figures 1a, 1d, 2d, 7b. In some experiments the siLUC recovers to 100%, in others this is 80%, or 180%. For instance, siUBN2 shows a recovery to 100%, which is considered a defect (Figure 7b, with the control showing an overshoot to 180%). At the same time, siH3.3 recovers to 80% (as does siLUC) in Figure 1d, which is considered normal recovery. To what extent does this variation between experiments affect the conclusions?

7. The levels of RNAPII and Ser2-P-modified RNAPII in Fig 3c do not seem to match (particularly for siVCP#2 at 8 hours after UV).

8. How much overlap is there between the UV-repressed genes defined in Fig 4b and the ATF3-response genes identified by the Egly lab (Epanchintsev et al. Mol Cell. 2017)?

9. Page 11 / Fig 5c: "HIRA knock-down not only impairs transcription recovery but also mitigates repression of short genes". Could the authors speculate on how this could work? Related to this: a recent study suggested that repression of short genes after UV requires the ubiquitylation of RNAPII at RPB1-K1268 and subsequent degradation (Tufegdžić Vidaković et al. Cell. 2020). It might be helpful to comment on this and mention that siHIRA does not interfere with RNAPII degradation (Fig 3d), but does have an impact on the repression of short genes (likely through ATF3).

10. Why do ATF3 levels in Fig 6e not drop at 24 hours after UV in siLUC cells (as they do in Fig 6b)?

11. The western blot in Fig S3a (middle one) should be replaced with a less saturated version.

Textual suggestions

12. Page 8: No hyphen between western blot.

13. Page 8: "abolished in TCR-deficient cells as expected^{26,65}". I believe citing the following studies would also be appropriate here: Nakazawa et al. Nat Gen. 2012; Bregman et al. PNAS. 1996.

14. Transcription-Coupled Repair (hyphenate throughout text)

15. Also mention somewhere that ERCC6 = CSB

Reviewer #3 (Remarks to the Author):

In this MS, the authors present evidence for a global role for histone chaperone HIRA in transcription re-start after UV induced DNA damage. This function depends on UBN2 and inhibition of ATF3 but is independent of histone H3.3 deposition. This is an interesting and important study, because it changes our perspective on the role of HIRA in transcription re-start after DNA damage. The data in support of H3.3 independent global transcription re-start is mostly convincing, pending addressing some technical concerns in points 1-3 below. This is an important contribution in its own right. The specific function of ATF3 is not defined here, but legitimately not the focus of this MS. The function of UBN2 is not defined and is an area where more mechanistic insight would be good if possible, but a detailed understanding beyond the scope of this MS.

1. Figure 1A appears to show that only HIRA KD suppresses EU nuclear levels – no effect of UBN1 and CABIN1 KD. This is key data for the conclusion that HIRA is required independent of H3.3 deposition. For this conclusion, it is important that the western blot is performed at the 24hr timepoint when EU levels are assayed. Rates of HIRA, UBN1, CABIN1 protein recoveries may differ. The authors should confirm this and/or perform western blot at the appropriate time.

2. Similarly the H3.3 deposition assays in Figure 1B are performed at 2hrs, not 24hrs. Differences in rates of protein and H3.3 recovery after KD could account for the differences at 24hrs. At a minimum, assays should be performed at the same time point and ideally more comprehensive time courses are required.

3. In relation to above, the H3.3 deposition assays are particularly important. For example, UBN1 might be in large excess over HIRA. Therefore knock down of the former might have no effect relative to the latter. It is important to show that H3.3 deposition is blocked at the same time that transcription is resumed.

4. A caveat to the interpretation that HIRA is required for restart of genes that are not damaged is that the regulatory elements, but not the genes, are damaged. The authors should discuss.

5. The data in Supp Fig 5E showing that HIRA binds to the ATF3 is not very convincing. Does HIRA binding to promoters/genes correlate with gene expression across the whole genome? Have the authors confirmed with a 2nd antibody? Alternatively, direct regulation of ATF3 by HIRA is not critical for the MS, so the authors could remove.

6. Figure 6F should also show ATF3 transcript levels after ASF1a KD.

7. In Fig S5F, the overlap of HIRA-regulated and ATF3 bound genes seems very substantial. However, the authors should calculate the fold enrichment over random increase and the associated p value.

8. In Figure 7A, the authors show that knock down of UBN2 suppresses deposition of H3.3. Is it surprising that UBN2 is not redundant with UBN1?

9. In Figure 7F, the authors propose an ATF3-independent local function for HIRA/H3.3 in transcription recovery. This predicts that after KD of HIRA and ATF3, as in Figure 6G, local transcription recovery is still inhibited. This seems like a key experiment, which I do not see.

Point-by-point responses to reviewers' comments:

We thank all three reviewers for their thorough evaluation of our manuscript and for constructive criticism. Their insightful comments helped to improve our manuscript and strengthen our conclusions. The main additions to our work are as follows:

- We consolidate **the roles of UBN2 and ASF1B in transcription recovery** following UVC damage
- We include **additional controls for nascent transcript imaging**: we provide unnormalized data, we monitor new H3.3 deposition and the protein levels of HIRA complex subunits at the time of transcription recovery, and we validate data by qRT-PCR.
- We provide a **revised final model**, which better reflects our conclusions and includes HIRA connection to global genome NER.

Reviewer #1:

This manuscript reports a requirement for HIRA, as well as ASF1B and UBN2, in transcription restart after UV damage repair, but is independent of UBN1 or ASF1A. They also report that loss of HIRA or ASF1B leads to elevated levels of the transcriptional repressor ATF3 which partly accounts for the failure to restart transcription after UV repair. In general the experiments are well designed and the story is interesting, although there are a few inconsistencies and gaps that need filling to make a coherent story. They have not convincingly shown that this is multiple pathways, nor involves a non-canonical pathway, as opposed to one canonical pathway for HIRA.

Major:

- All of the transcription data throughout the paper are normalized to 100% for each cell line in the absence of DNA damage. It is possible that depletion of HIRA *per se* leads to a global increase in transcription regardless of DNA damage. It is important that they show at least some experiments without normalization to 100% so the global effects on transcription before UV damage can be assessed, as it will affect the interpretation of all the data.

We indeed normalize the transcription data to undamaged cells to account for potential variations in basal transcription levels between conditions, allowing us to assess if the initial transcription levels are restored 24h post damage. Depletion of HIRA *per se* does not lead to an increase in transcription in the absence of DNA damage but rather to a moderate decrease in transcription (about 30%, as can be seen on the microscopy images Fig. 1a, and consistent with our previous observations, Adam et al, Cell 2013). For clarity, we now show transcription data without normalization for HIRA, UBN1, CABIN1 and H3.3 knockdowns (Suppl. Fig. 1b).

- It is concluded that new H3.3 is not required for transcriptional restart after UV damage, however this is not convincing because the depletion of H3.3 is not complete in

Fig. 1D. This caveat needs to be mentioned in the text, because the remaining H3.3 may be sufficient for transcriptional restart.

The conclusion is about new H3.3 histones because the depletion of newly synthesized H3.3 is nearly complete, as assessed by imaging (Fig.1c) and as expected, this does not lead to a complete loss of total H3.3 levels (western blot Fig. 1d) because new histones represent only a fraction of total histones.

Similarly, in the experiment in Fig. D, the extent of transcriptional repression upon UV damage is a lot less than in other experiments, going down to about 65% vs 40% in all the other experiments. This difference seems to be attempted to be hidden by the altered scale on the vertical axis. All the EU fluorescence graphs should be shown with an axis from 0 to 100% so as not to infer a greater effect, which is what happens when one starts the axis at 20% or 40%, as they do in this study.

The extent of transcriptional repression is not as pronounced in Fig. 1d most likely because the experimental set up is different for H3.3 knockdown: cells are irradiated 24h post siRNA instead of 48h for HIRA complex subunit knockdown because long-term H3.3 depletion was not well tolerated by the cells. For consistency, we now show all the graphs with the same scale on the vertical axis. We chose to start at 20% rather than 0% for a better visualization of transcription inhibition and recovery.

- They conclude that loss of HIRA leads to elevated levels of the ATF3 transcriptional repressor. That is shown in Fig. 6A and D and H at 24 hours, but the levels of ATF3 are actually reduced in westerns Fig. 6G and Fig. 7D upon HIRA depletion. Consistent results need to be shown.

We reproducibly observe elevated ATF3 levels in HIRA-knocked down cells 24h post UV as shown both by western blot (Fig. 6a, 6d) and imaging (Fig. 6b, 6h). Samples for the western blots shown in original Fig. 6g and 7d were prepared from unirradiated cells, which we now specify on Fig. 7d, and longer exposures are thus shown for ATF3. ATF3 levels are not increased, and even slightly reduced, upon HIRA knockdown in undamaged cells, consistent with what is shown in Fig. 6a, 6b, and 6d at the 0h time point. Twenty-four hours post irradiation in contrast, ATF3 levels are massively increased in HIRA-knocked down cells. To illustrate this, we replaced the western blot of Fig. 6g (now Fig. 6h) by a western blot performed 24h post UV, which shows elevated ATF3 upon HIRA knockdown.

- In Fig. 6A and D, there are much more increase of ATF3 protein levels at 24 hours than 8 hours in siHIRA cells. However, nascent transcription level in siHIRA cells at 24 hours is higher than that at 8 hours in Fig. 6G and all other figures, which is not consistent with the idea that ATF3 represses transcription after damage.

It is true that transcription recovery is observed with higher ATF3 levels at 24h than at earlier time points post UV. This is not inconsistent with the role of ATF3 is controlling transcription restart. ATF3 does not repress transcription at early time points after damage but impairs transcription restart at late time points. Transcription inhibition at early time points occurs via ATF3-independent mechanisms including RNAPII stalling on UV lesions.

- What do they speculate to be the molecular mechanism whereby loss of ASF1B or HIRA leads to elevated ATF3 transcript levels 24 hours after UV damage?

Our data supports the hypothesis of a silencing of ATF3 transcription by HIRA and ASF1B, which could involve the recruitment of transcriptional repressors to the ATF3 gene as discussed on p.18 of our revised manuscript.

- Given that HIRA depletion leads to loss of UBN2 protein, the results reported for HIRA could be indirect effects that are a consequence of loss of UBN2. This could be tested by seeing if reexpression of UBN2 is sufficient to restore transcription restart in a HIRA depleted cell.

We thank the reviewer for this interesting suggestion. We succeeded to re-express, at least partially, UBN2 in HIRA-depleted cells by performing dual transfection with siRNA and plasmid DNA as shown on the western blot below (empty: empty plasmid; UBN2: UBN2 plasmid obtained from GenScript).

However, this dual transfection protocol was quite cytotoxic and thus turned out to be difficult to combine with the labeling of nascent transcripts. We ended up with very low levels of nascent transcript labeling (3-4 times lower signal than usual), which was not reliable and prevented us from drawing solid conclusions regarding the potential effect of UBN2 re-expression on transcription recovery post UV damage. Therefore, we chose not to include those experiments in the manuscript.

- Does loss of UBN2 also lead to elevated ATF3 protein levels?

Loss of UBN2 does not lead to elevated ATF3 protein levels as shown in the original version of our manuscript (Fig. 7c) and now confirmed using a second siRNA against UBN2 (Suppl. Fig. 6c), which similarly impairs transcription recovery (Suppl. Fig. 6b). This is why we conclude that ATF3 and UBN2 operate in independent pathways.

- Similarly they show that HIRA depletion does not affect ASF1A levels, but they need to show whether this is the case with ASF1B or not.

In Suppl. Fig. 1d (western blot panel), we show that ASF1B levels display a very modest

decrease upon HIRA depletion and below is another experiment leading to the same conclusion. HIRA depletion thus does not lead to loss of ASF1B (p.13 of our revised manuscript).

- It is a bit of a stretch to say that these are non canonical roles for HIRA, given that they did not rule out a role for parental H3.3 incorporation, only new H3.3. Nor did they rule out a role for UBN2 or ASF1B in ATF3 expression, which if they were involved would make it a canonical role for HIRA. As such all references to a non canonical roles for HIRA should be removed.

We actually rule out a role for UBN2 in controlling ATF3 levels (Fig. 7c and Suppl. Fig. 6c). We do not discard the possibility that parental H3.3 deposition by the HIRA complex can play a role in transcription restart post UV damage, as explained in our discussion (p.17), but parental H3.3 deposition by HIRA relies on both ASF1A and ASF1B and does not involve UBN2 (Torné et al, NSMB 2020) while in our case only ASF1B contributes and UBN2 is involved. This is why we claim that HIRA function in transcription restart post UV is non-canonical because it is distinct from hitherto reported functions of HIRA. Furthermore, it does not involve the known function of HIRA at UV damage sites, namely new H3.3 deposition.

- The model figure shows new H3.3 promoting transcription restart but that is the opposite of what is concluded in the paper.

We have revised the model to better reflect our conclusions (Fig. 7f).

- Their model shows that ASF1B and HIRA and UBN2 function together to promote transcription restart. However, they have not shown a requirement for ASF1B in transcription restart after UV damage. It is critical that they examine whether ASF1B is required for transcriptional restart or not after UV damage.

This is true. We now examine the impact of ASF1A and ASF1B knockdowns on transcription restart after UV damage and show that only ASF1B knockdown impairs transcription restart (Fig. 6g, Suppl. Fig. 5g and manuscript p.13).

- The data shown with the EU incorporation measured by fluorescence throughout the paper in the absence of HIRA clearly show a profound defect in transcriptional restart. However, that is not apparent in the BrU seq analyses in Fig. 4D and Fig. 5D, where the defect without HIRA appears quite subtle compared to siLUC. The authors need to

explain the discrepancy.

The data in Fig. 4d and 5d is presented in heatmaps as log₂ fold change, which may give the impression of smaller differences. The heatmaps still show a marked defect in transcription restart for most genes at the 24.5 h time point (more yellow genes in siLUC vs. dark blue genes in siHIRA).

- It is possible that all the factors shown function in the same pathway to promote transcription restart after UV, as this has not been addressed. This seems particularly likely given that depletion of UBN2 is equally defective as depletion of HIRA for transcriptional restart. Does depletion of ATF3 also partially reverse the transcription restart defect seen upon UBN2 depletion?

We do show that UBN2 and ATF3 operate in independent pathways for transcription restart since UBN2 knockdown does not affect ATF3 levels and reciprocally (Fig. 7c-d and Suppl. Fig. 6c). Given that ATF3 levels are not elevated upon siUBN2, we did not attempt to rescue transcription restart by knocking down ATF3 in UBN2-depleted cells as we did in HIRA-depleted cells.

Minor:

- Fig. S4d mRNA levels cannot reflect nascent RNA production even of short half-life transcripts. So, the authors should use intron-exon junction primers to validate the results from BrU-seq.

We agree with the reviewer that mRNA levels of short half-life transcripts do not strictly correspond to nascent RNA levels. Therefore, we now refer to 'Transcript levels' rather than 'Nascent transcription' on the RT-qPCR graphs in Suppl. Fig. 4e. We chose to focus on short half-life transcripts instead of using intron-exon junction primers because UV irradiation impacts splicing (Munoz et al, Cell 2009). Furthermore, we can also expect some effect of HIRA on splicing via H3.3 deposition because the H3.3K36me₃ reader ZMYND11 regulates intron retention (Guo, Mol Cell 2014). If splicing is affected, introns may be retained in the mature RNA and their presence does not necessarily reflect nascent RNA. To confirm that examining short half-life transcripts with exonic primers was a good proxy for nascent RNA production, we performed RT-qPCR with the same primers on nascent RNA (BrdU-labeled RNA isolated on magnetic beads coated with an anti-BrdU antibody). As shown below, similar trends were obtained using both approaches, with exacerbated differences between siLUC and siHIRA conditions when running RT-qPCR on nascent RNA. We have included in Fig. 4f the RT-qPCR result on BrU-labeled RNA compared to BrU-seq data for the RAD50 gene product.

• Fig. 5c, the authors claim that “HIRA-knock-down not only impairs transcription recovery but also mitigates transcription repression of short genes”, which might not be true. Because there are much higher basal transcription levels in siHIRA cells compared to controls at earlier time points.

We did not observe higher basal transcription levels in siHIRA cells but rather the opposite (30% reduction) as shown in Suppl. Fig. 1b. Furthermore, data shown in Fig. 5c are normalized to undamaged conditions and short genes show increased transcript levels at early time points (0.5h post UV) in HIRA-depleted cells. This is why we conclude that HIRA knock-down mitigates transcription repression of short genes.

Reviewer #2:

The manuscript by Bouvier et al. from the Polo lab dissects regulatory pathways by which the histone chaperone HIRA regulates transcription recovery following DNA damage repair. This is a thoughtful and elegant follow-up of previous work from the Almouzni lab showing that HIRA is involved in transcription recovery, and implying that this involved new histone H3.3 deposition acting as an early bookmarking mechanisms to restart transcription later on (Adam et al. Cell. 2013).

The current study shows that the deposition of H3.3 is not involved in transcription recovery. Rather HIRA regulates transcription restart by two distinct mechanisms: (1) shutting down ATF3 transcription at late time-points after UV (24 hrs), to relieve its repressive impact and enable cells to re-initiate transcription, and (2) stabilize the protein levels of regulatory subunit UBN2. The study provides important insights into how cells recover transcription after DNA damage repair is completed by revealing that HIRA regulates ATF3 expression at the transcriptional level. This reveals an additional regulatory layer by showing that DNA repair itself is not sufficient to restart transcription.

The study does not provide mechanistic insight into how the HIRA/ATF3-dependent pathway works, and in particular how the HIRA/UBN2/ATF3-independent pathway operates. The regulation of ATF3 expression in response to UV damage is complex and regulated at multiple levels. Fully understanding his regulatory network will require more follow-up work. However, this study is a first important step to better understand the mechanisms involved.

The manuscript is logical and well-written, the figures are very clear, and the experiments are thoughtful. This work deserves being considered in Nature Communications, should the authors appropriately address the following points:

Major Points

1. The recruitment of HIRA to sites of local UV damage was previously shown to depend on global genome repair (GGR) factor DDB2 (Adam et al. Cell. 2013; Fig S5A). This links HIRA recruitment to GGR rather than to TCR. The role of VCP in recruiting HIRA would also fit with this scenario, considering that VCP is recruited by and involved in handling ubiquitylated DDB2 at sites of local damage (Puumalainen et al. Nat Comms. 2014, citation 63 in this study). I am not convinced that the recruitment of HIRA is linked to VCP-dependent extraction of RNAPII as suggested by the model depicted in Fig 7f. After all, HIRA recruitment to sites of local UV damage is independent of transcription (Adam et al. Cell. 2013). The authors should address whether H3.3 deposition is dependent on DDB2 (using siRNAs against DDB2 as done in Adam et al. Cell. 2013 (Fig S5A)). It seems as if the authors are deliberately vague on this issue (“HIRA is recruited to damaged chromatin regions in a ubiquitin-dependent manner”).

We fully agree that HIRA recruitment to UV-damaged chromatin is linked to GGR. Supporting this idea, we now show that new H3.3 deposition at UV sites is dependent on the GGR factor DDB2 (Suppl. Fig. 2c). Even though GGR plays a major part in recruiting

HIRA, we cannot rule out that HIRA is not also recruited by ubiquitylation events within the TCR pathway because TCR represents only a minor fraction of NER. Still, we have revised our model (Fig. 7f) not to suggest that HIRA is solely recruited by ubiquitylated RNAPII.

If DDB2 regulates H3.3 deposition, it would strengthen the paper to functionally link HIRA and H3.3 function to GGR, for instance by a UDS assay in wild-type cells (including DDB2 as a control), and revise the model in Fig 7f accordingly. It is possible that the local role of HIRA stimulates GGR, while it has a global impact downstream from TCR in regulating transcription recovery.

HIRA has no impact on GGR *per se*, as previously shown by measuring repair synthesis at sites of local UV damage (Adam et al, Cell 2013) so the function of HIRA-mediated H3.3 deposition in the GGR pathway is still an open issue as discussed p.17 of our revised manuscript.

2. I applaud the use of the biological scaling normalization (in this case RRS experiments) to normalize the BrU-seq data. The validation by qRT-PCR of several genes (Fig S4d) makes a strong case that this approach is valid and matches the normalized BrU-seq data very well. I suggest to show the BrU-seq and qRT-PCR data for one gene next to each other in the main figure. The Bru-seq data also contains information on the distribution of nascent transcripts within the genome, which is not really used in this study. It could be informative to show the BrU-seq read distribution from siLUC and siHIRA cells mapped on the genome for at least one gene at different time-points after UV as an example (e.g. RAD50 or NIPBL).

Based on a comment by Reviewer#1, we performed RT-qPCR validation experiments also on nascent RNA (BrU-labeled RNA isolated on magnetic beads coated with an anti-BrdU antibody). We have included in main Fig. 4f the RT-qPCR result on BrU-labeled RNA compared to BrU-seq data for the RAD50 gene product.

We now show the BrU-seq read distribution mapped on the NIPBL gene at different time points after UV in siLUC and siHIRA cells (Suppl. Fig. 4a). The distribution of BrU-seq reads show a similar profile in control and HIRA-depleted cells with an accumulation of reads in the promoter-proximal region early after UV followed by release from promoter-proximal pausing. However, at late time points post UV, transcript levels do not reach back pre-irradiation levels in HIRA-depleted cells. These results strongly suggest that HIRA is not involved in release from promoter-proximal pausing but plays a downstream role in transcription recovery (p.9-10) of our revised manuscript).

3. The authors argue that ASF1B cooperates with HIRA in both pathways (HIRA-ASF1B-ATF3 and HIRA-ASF1B-UBN2). They show in Fig 6e that knockdown of ASF1B affects ATF3 proteins levels, while ASF1A knockdown does not. The authors should test if ASF1B knockdown leads to defective transcription recovery, while siASF1A does not.

We now examine the impact of ASF1A and ASF1B knockdowns on transcription restart after UV damage and show that only ASF1B knockdown impairs transcription restart (Fig. 6g and manuscript p.13).

4. The partial restoration of transcription recovery measured by BrU incorporation in Fig

6g in siHIRA+siATF3 cells compared to siHIRA cells is very elegant. However, in this experiment there appears to be a partial recovery in siHIRA cells, which is not observed in other experiments (Fig 1a, Fig 2d). Is this also seen under optimized conditions that show a stronger defect of siHIRA? The authors should also monitor the expression of an ATF3 target gene by qRT-PCR (e.g. NIPBL or RAD50) under siHIRA versus siHIRA+siATF3 conditions to see whether a (partial) rescue is observed. This would considerably strengthen the RRS data in Fig 6g.

HIRA knockdown is efficient in those experiments as shown on the western blot panel (Fig. 6h). We suspect that the transcription recovery defect is not that pronounced because of the double knockdown conditions used in those experiments, RRS analyses being very sensitive to cell state.

To strengthen these results, following the suggestion of the reviewer, we monitored the transcript levels of the ATF3 target gene RAD50 by RT-qPCR and observed a rescue in siHIRA+siATF3 conditions. These new results are included in Fig. 6i.

5. The authors should speculate in the discussion how they think HIRA might regulate ATF3 transcription after UV? Fig S5e shows published ChIP-seq data showing possible HIRA binding to the ATF3 gene promoter and nearby regulatory elements (in unirradiated cells). However, ATF3 expression is not increased in unirradiated cells depleted of HIRA (Fig 6b), meaning that loss of HIRA only leads to elevated ATF3 levels (transcript and protein) after UV irradiation. The authors are invited to include ideas / speculation on how this could work in the discussion.

We speculate that HIRA, together with ASF1B, might repress ATF3 transcription post UV by binding to regulatory sequences of the ATF3 gene and recruiting transcriptional repressors (discussed p.18 of our revised manuscript). ATF3 expression is not increased in unirradiated cells depleted of HIRA because ATF3 expression is stress-induced so the effect of HIRA depletion on ATF3 levels is detectable only after UV irradiation. Moreover, we observed increased ASF1B levels post UV (Suppl. Fig. 5f), which may further enhance the silencing of ATF3 after UV damage.

Minor Points

6. There is considerable variation in the Recovery of RNA synthesis (RRS) experiments when comparing Figures 1a, 1d, 2d, 7b. In some experiments the siLUC recovers to 100%, in others this is 80%, or 180%. For instance, siUBN2 shows a recovery to 100%, which is considered a defect (Figure 7b, with the control showing an overshoot to 180%). At the same time, siH3.3 recovers to 80% (as does siLUC) in Figure 1d, which is considered normal recovery. To what extent does this variation between experiments affect the conclusions?

Those experiments are very sensitive to cell state, which explains some of the variability, and results may also vary depending on the use of EU or BrU to label nascent RNA. This is why we always compare the recovery profiles to siLUC in the same experiment and we run those experiments multiple times to be able to draw solid conclusions.

7. The levels of RNAPII and Ser2-P-modified RNAPII in Fig 3c do not seem to match (particularly for siVCP#2 at 8 hours after UV).

There is no perfect match because the antibody against total RNAPII gives quite a poor signal. The result was not improved after reloading these samples. However, the quantifications are run on the RNAPII Ser2-P signal, which is much more robust.

8. How much overlap is there between the UV-repressed genes defined in Fig 4b and the ATF3-response genes identified by the Egly lab (Epanchintsev et al. Mol Cell. 2017)?

62% of UV-repressed genes and 64% the genes that require HIRA for transcription recovery were identified as ATF3-bound post UVC by the Egly lab. We now mention this in our revised manuscript (p.13).

9. Page 11 / Fig 5c: “HIRA knock-down not only impairs transcription recovery but also mitigates repression of short genes”. Could the authors speculate on how this could work? Related to this: a recent study suggested that repression of short genes after UV requires the ubiquitylation of RNAPII at RPB1-K1268 and subsequent degradation (Tufegdžić Vidaković et al. Cell. 2020). It might be helpful to comment on this and mention that siHIRA does not interfere with RNAPII degradation (Fig 3d), but does have an impact on the repression of short genes (likely through ATF3).

This is an interesting point that we now discuss p.17-18 of our revised manuscript. We do not think that the impact of HIRA on the silencing of short genes is mediated by ATF3 because ATF3 levels rise much later. ATF3 does not repress transcription at early time points after damage but impairs transcription restart at late time points.

10. Why do ATF3 levels in Fig 6e not drop at 24 hours after UV in siLUC cells (as they do in Fig 6b)?

This is due to the fact that cells were harvested 48h instead of 72h post siRNA. Below, we show a similar experiment 72h post siRNA where ATF3 levels go down 24h post UV. We reach the same conclusion: ASF1B knockdown leads to elevated ATF3 levels.

11. The western blot in Fig S3a (middle one) should be replaced with a less saturated version.

We have replaced the middle western blot in Suppl. Fig. 3a by another loading of the same samples, which was less saturated, and we have also replaced the top western blot by a less saturated version.

Textual suggestions

12. Page 8: No hyphen between western blot.

Corrected everywhere in the manuscript.

13. Page 8: “abolished in TCR-deficient cells as expected^{26,65}”. I believe citing the following studies would also be appropriate here: Nakazawa et al. Nat Gen. 2012; Bregman et al. PNAS. 1996.

We did not cite these studies because they do not describe RNAPII degradation following ubiquitylation post UV.

14. Transcription-Coupled Repair (hyphenate throughout text)

Done

15. Also mention somewhere that ERCC6 = CSB

Added on p.7 of our revised manuscript

Reviewer #3:

In this MS, the authors present evidence for a global role for histone chaperone HIRA in transcription re-start after UV induced DNA damage. This function depends on UBN2 and inhibition of ATF3 but is independent of histone H3.3 deposition. This is an interesting and important study, because it changes our perspective on the role of HIRA in transcription re-start after DNA damage. The data in support of H3.3 independent global transcription re-start is mostly convincing, pending addressing some technical concerns in points 1-3 below. This is an important contribution in its own right. The specific function of ATF3 is not defined here, but legitimately not the focus of this MS. The function of UBN2 is not defined and is an area where more mechanistic insight would be good if possible, but a detailed understanding beyond the scope of this MS.

1. Figure 1A appears to show that only HIRA KD suppresses EU nuclear levels – no effect of UBN1 and CABIN1 KD. This is key data for the conclusion that HIRA is required independent of H3.3 deposition. For this conclusion, it is important that the western blot is performed at the 24hr timepoint when EU levels are assayed. Rates of HIRA, UBN1, CABIN1 protein recoveries may differ. The authors should confirm this and/or perform western blot at the appropriate time.

In Fig.1a, western blot samples were prepared at the time of UV irradiation, which we now specify in the corresponding figure legend. In addition, we have now included another western blot with samples prepared 24h later (Suppl. Fig. 1c). At this time point, the impact of UBN1 and CABIN1 knockdowns on HIRA levels is slightly more pronounced but still not comparable to a knockdown of HIRA.

2. Similarly the H3.3 deposition assays in Figure 1B are performed at 2hrs, not 24hrs. Differences in rates of protein and H3.3 recovery after KD could account for the differences at 24hrs. At a minimum, assays should be performed at the same time point and ideally more comprehensive time courses are required.

New H3.3 deposition assays were performed 2h instead of 24h post UV because new H3.3 accumulate at UV sites only around the time of UV irradiation. To assess the status of newly deposited H3.3 24h post UV, i.e. at the time of transcription restart, we labeled new histones immediately after UV irradiation and fixed the cells 24h later. The levels of new H3.3 at UV sites were reduced in HIRA-, UBN1-, CABIN1- and H3.3-knocked down cells as observed 2h post UV (24h graphs in Fig. 1b-c).

3. In relation to above, the H3.3 deposition assays are particularly important. For example, UBN1 might be in large excess over HIRA. Therefore knock down of the former might have no effect relative to the latter. It is important to show that H3.3 deposition is blocked at the same time that transcription is resumed.

Please see our response to the above comment.

4. A caveat to the interpretation that HIRA is required for restart of genes that are not damaged is that the regulatory elements, but not the genes, are damaged. The authors should discuss.

This is a good point, which we have included in our discussion (p.18).

5. The data in Supp Fig 5E showing that HIRA binds to the ATF3 is not very convincing. Does HIRA binding to promoters/genes correlate with gene expression across the whole genome? Have the authors confirmed with a 2nd antibody? Alternatively, direct regulation of ATF3 by HIRA is not critical for the MS, so the authors could remove.

The data in Suppl. Fig 5e is ChIP-seq data from Pchelintsev et al, Cell Rep 2013. It is only intended to show that a low level of HIRA is detectable on regulatory regions of the ATF3 gene in undamaged conditions leading us to speculate that HIRA may silence ATF3 transcription post UV by recruiting transcriptional repressors to the ATF3 gene.

6. Figure 6F should also show ATF3 transcript levels after ASF1a KD.

We have included in Fig. 6f data showing that ATF3 transcript levels are not significantly upregulated 24h post UV irradiation in ASF1A-knocked down cells, which is consistent with our observations at the protein level.

7. In Fig S5F, the overlap of HIRA-regulated and ATF3 bound genes seems very substantial. However, the authors should calculate the fold enrichment over random increase and the associated p value.

There is a significant association between HIRA-regulated and ATF3-bound genes as shown by Fisher's exact test: odds ratio= 2.078, $p < 2.2e^{-16}$ (now indicated in Suppl. Fig. 5h).

8. In Figure 7A, the authors show that knock down of UBN2 suppresses deposition of H3.3. Is it surprising that UBN2 is not redundant with UBN1?

Both UBN1 and UBN2 knockdowns do not fully suppress but partially reduce new H3.3 deposition (about 50% reduction compared to control as shown on Fig. 1b and 7a). Additionally, we noticed that combined depletion of UBN1 and UBN2 had a greater effect (Suppl. Fig. 6a), arguing that both subunits have additive rather than redundant roles. We now mention this point in our manuscript (p.14-15).

9. In Figure 7F, the authors propose an ATF3-independent local function for HIRA/H3.3 in transcription recovery. This predicts that after KD of HIRA and ATF3, as in Figure 6G, local transcription recovery is still inhibited. This seems like a key experiment, which I do not see.

Based on comments by Reviewers #1 and #2, we have revised our final model because we had no data directly supporting a local function of HIRA in transcription recovery. The local function of HIRA at UV sites relates not only to Transcription-Coupled Repair but also to Global Genome Repair and involves new H3.3 deposition.

REVIEWERS' COMMENTS

Reviewer #1 (Remarks to the Author):

I am satisfied with the authors response to my comments. This is a nice quality, thorough and insightful study

Reviewer #2 (Remarks to the Author):

The revised manuscript by Boivier et al. from the lab of Sophie Polo lab has done an excellent job at addressing all my points.

Showing the RT-qPCR and BrU-seq data next to each other (Fig 4f) and showing the rescue of RAD50 transcript levels by RT-PCR in Fig 6i strengthens the conclusions.

I recommend publication of this manuscript in Nature Communications.

Suggestion for consideration by the authors:

- Page 7: DNA Damage-Binding protein 2 (with hyphen)
- Fig 3c, d: consider no showing unmodified RNAPII blots
- Fig 4 mentions BrU-seq while the manuscript refers to Bru-seq.

Reviewer #3 (Remarks to the Author):

The reviewers have addressed most of my comments. I still feel that Figure 5E is overstated - how do they know the region marked "enhancer" is an enhancer. Still this is a minor point.

Responses to reviewers' comments

We thank all three reviewers for their positive evaluation of our revised manuscript. We have included our responses below when further changes were suggested.

Reviewer #1 :

I am satisfied with the authors response to my comments. This is a nice quality, thorough and insightful study.

Reviewer #2 :

The revised manuscript by Bouvier et al. from the lab of Sophie Polo lab has done an excellent job at addressing all my points.

Showing the RT-qPCR and BrU-seq data next to each other (Fig 4f) and showing the rescue of RAD50 transcript levels by RT-PCR in Fig 6i strengthens the conclusions.

I recommend publication of this manuscript in Nature Communications.

Suggestion for consideration by the authors:

- Page 7: DNA Damage-Binding protein 2 (with hyphen)

Corrected

- Fig 3c, d: consider no showing unmodified RNAPII blots

We decided to keep those blots in even though we acknowledge that they are not perfect.

- Fig 4 mentions BrU-seq while the manuscript refers to Bru-seq.

Corrected in Fig 4a, 4f and Suppl. Fig. 4e

Reviewer #3:

The reviewers have addressed most of my comments. I still feel that Figure 5E is overstated - how do they know the region marked "enhancer" is an enhancer. Still this is a minor point.

We modified the corresponding panel in Suppl. Fig. 5e by labeling regulatory regions without specifying promoter or enhancer.